# SLC6A20 transporter: a novel regulator of brain glycine homeostasis and NMDAR function

Mihyun Bae[1,†], Junyeop Daniel Roh[2,†], Youjoung Kim[2], Seong Soon Kim[3], Hye Min Han[4], Esther Yang[5], Hyojin Kang[6], Suho Lee[1], Jin Yong Kim[5], Ryeonghwa Kang[2], Hwajin Jung[1], Taesun Yoo[1], Hyosang Kim[2], Doyoun Kim[1] [iD], Heejeong Oh[2], Sungwook Han[2], Dayeon Kim[7], Jinju Han[7] [iD], Yong Chul Bae[4] [iD], Hyun Kim[5] [iD], Sunjoo Ahn[3], Andrew M Chan[8] [iD], Daeyoup Lee[2], Jin Woo Kim[2] [iD] & Eunjoon Kim[1,2,*] [iD]

## Abstract

Glycine transporters (GlyT1 and GlyT2) that regulate levels of brain glycine, an inhibitory neurotransmitter with co-agonist activity for NMDA receptors (NMDARs), have been considered to be important targets for the treatment of brain disorders with suppressed NMDAR function such as schizophrenia. However, it remains unclear whether other amino acid transporters expressed in the brain can also regulate brain glycine levels and NMDAR function. Here, we report that SLC6A20A, an amino acid transporter known to transport proline based on *in vitro* data but is understudied in the brain, regulates proline and glycine levels and NMDAR function in the mouse brain. SLC6A20A transcript and protein levels were abnormally increased in mice carrying a mutant PTEN protein lacking the C terminus through enhanced β-catenin binding to the *Slc6a20a* gene. These mice displayed reduced extracellular levels of brain proline and glycine and decreased NMDAR currents. Elevating glycine levels back to normal ranges by antisense oligonucleotide-induced SLC6A20 knockdown, or the competitive GlyT1 antagonist sarcosine, normalized NMDAR currents and repetitive climbing behavior observed in these mice. Conversely, mice lacking SLC6A20A displayed increased extracellular glycine levels and NMDAR currents. Lastly, both mouse and human SLC6A20 proteins mediated proline and glycine transports, and SLC6A20 proteins could be detected in human neurons. These results suggest that SLC6A20 regulates proline and glycine homeostasis in the brain and that SLC6A20 inhibition has therapeutic potential for brain disorders involving NMDAR hypofunction.

**Keywords** glycine transporter; neuropsychiatric disorders; NMDA receptor; PTEN; Slc6a20

**Subject Category** Neuroscience

## Introduction

Glycine, a well-known inhibitory neurotransmitter for glycine receptors in the nervous system, is also one of the obligatory co-agonists of *N*-methyl-D-aspartate (NMDA) receptors. Glycine and/or D-serine binds to the strychnine-insensitive site on the GluN1 subunit of the NMDARs, permitting subsequent binding of glutamate (Kuryatov *et al*, 1994; Schell *et al*, 1995; Mothet *et al*, 2000). Two glycine transporters, GlyT1 and GlyT2, have been identified as important regulators of glycine homeostasis in the brain (Smith *et al*, 1992; Liu *et al*, 1993a). GlyT1 is a Na²⁺-dependent high-affinity glycine transporter with an expression pattern in the brain that mirrors the distribution of NMDA receptors (Cubelos *et al*, 2005). Persistent hypofunction of GlyT1 (Tsai *et al*, 2004b; Gabernet *et al*, 2005) or acute treatment with the potent GlyT1 antagonists, such as sarcosine and *N*-[3-(4-fluorophenyl)-3-(4-phenylphenoxy)] propylsarcosine (NFPS), has been shown to enhance NMDAR currents and LTP in the hippocampus (Martina *et al*, 2004; Manahan-Vaughan *et al*, 2008). Naturally, glycine transporters have been suggested as novel therapeutic targets for brain disorders involving NMDAR dysfunction such as schizophrenia, alcohol dependence, and pain (Tsai *et al*, 2004a; Lane *et al*, 2006; Javitt, 2008; Harvey & Yee, 2013). However, whether other types of transporters are involved in the regulation of glycine homeostasis in the brain remains unclear.

We began the current study by characterizing a previously reported mouse line that lacks the C-terminal tail of the PTEN

1 Center for Synaptic Brain Dysfunctions, Institute for Basic Science (IBS), Daejeon, Korea
2 Department of Biological Sciences, Korea Advanced Institute for Science and Technology (KAIST), Daejeon, Korea
3 Therapeutics and Biotechnology Division, Korea Research Institute of Chemical Technology (KRICT), Daejeon, Korea
4 Department of Anatomy and Neurobiology, School of Dentistry, Kyungpook National University, Daegu, Korea
5 Department of Anatomy and Division of Brain Korea 21, Biomedical Science, College of Medicine, Korea University, Seoul, Korea
6 Division of National Supercomputing, KISTI, Daejeon, Korea
7 Graduate School of Medical Science and Engineering, KAIST, Daejeon, Korea
8 School of Biomedical Sciences, The Chinese University of Hong Kong, Hong Kong, Hong Kong SAR, China
*Corresponding author. Tel: +82 42 350 2633; Fax: +82 42 350 8127; E-mail: kime@kaist.ac.kr
†These authors contributed equally to this work

protein (Knafo *et al*, 2016), a lipid, and protein phosphatase that antagonizes the phosphoinositide 3-kinase (PI3K) pathway ($Pten^{\Delta C/\Delta C}$ mice) (Knafo *et al*, 2016). Our initial motivation to study PTEN was that it regulates brain development and synaptic/neuronal functions (Backman *et al*, 2001; Endersby & Baker, 2008) and is also implicated in brain disorders, including macrocephaly, epilepsy, Rett syndrome, and autism spectrum disorders (ASD) (Crespi *et al*, 2010; Tilot *et al*, 2016). Previous studies for brain PTEN functions, which used various mouse lines with *Pten* haploinsufficiency (Napoli *et al*, 2012; Clipperton-Allen & Page, 2014), neuronal *Pten* deletion (Suzuki *et al*, 2001; Kwon *et al*, 2006; Takeuchi *et al*, 2013; Williams *et al*, 2015; Cupolillo *et al*, 2016), and glial *Pten* deletion (Lugo *et al*, 2013; Lugo *et al*, 2014), have mainly focused on the phosphatase activity of PTEN.

We noted, however, that PTEN contains a C-terminal tail that contains a PDZ domain-binding (PB) motif that can bind various PDZ-containing proteins, including PSD-95 (Adey *et al*, 2000; Wu *et al*, 2000; Tolkacheva *et al*, 2001; Takahashi *et al*, 2006; Bonifant *et al*, 2007; Jurado *et al*, 2010), an abundant excitatory postsynaptic scaffolding protein (Sheng & Sala, 2001; Sheng & Hoogenraad, 2007; Sheng & Kim, 2011). Intriguingly, the PTEN-PB domain has been implicated in the regulation of excitatory synaptic strength during long-term depression (LTD) (Jurado *et al*, 2010; Knafo & Esteban, 2017) and (Aβ)-induced weakening of excitatory synapses in Alzheimer's disease (Knafo *et al*, 2016). However, these studies did not fully investigate other synaptic functions of the PTEN C terminus or non-synaptic PTEN functions such as nuclear localization and regulation of gene expression (Gil *et al*, 2007; Planchon *et al*, 2008; Howitt *et al*, 2012; Bassi *et al*, 2013; Zhang *et al*, 2013; Goh *et al*, 2014; Fricano-Kugler *et al*, 2018; Igarashi *et al*, 2018).

Our results indicated that $Pten^{\Delta C/\Delta C}$ mice show suppressed LTD, as reported previously (Jurado *et al*, 2010; Knafo *et al*, 2016; Knafo & Esteban, 2017), and unexpectedly suppressed long-term potentiation (LTP) through decreased NMDAR-mediated currents. While searching for the underlying mechanisms, we found that mRNA and protein levels of SLC6A20A, an amino acid transporter known to transport proline based on *in vitro* data (Smith *et al*, 1995; Nash *et al*, 1998; Kiss *et al*, 2002; Kowalczuk *et al*, 2005; Takanaga *et al*, 2005; Broer & Gether, 2012), were substantially increased, which was associated with abnormally decreased extracellular brain glycine levels. Elevating brain glycine levels back to a normal range by antisense oligonucleotides for SLC6A20A, or by the competitive GlyT1 antagonist sarcosine, normalized NMDAR function and repetitive climbing behavior in $Pten^{\Delta C/\Delta C}$ mice. Importantly, a new mouse line lacking SLC6A20A showed increased extracellular brain glycine levels and NMDAR function. These results suggest that SLC6A20 is a novel regulator of brain glycine levels and NMDAR function that has therapeutic potential for brain disorders involving suppressed NMDAR function.

# Results

## Characterization of $Pten^{\Delta C/\Delta C}$ mice

To explore the function of the C-terminal PDZ-binding motif of PTEN (PTEN-PB), we characterized a *Pten*-mutant mouse line that lacks the last five amino acid (aa) residues of PTEN (aa 399–403)

(Knafo *et al*, 2016) (Fig EV1A). These $Pten^{\Delta C/\Delta C}$ mice showed normal rates of birth, growth and survival, and gross morphology of the brain relative to wild-type (WT) mice, as shown by staining for NeuN (neuronal marker) (Fig EV1B and C).

$Pten^{\Delta C/\Delta C}$ mice also displayed normal levels of PTEN protein in the brain, as shown by immunoblotting using whole-brain and hippocampal samples (Fig EV1D–F). PTEN phosphorylation on Ser380 and Thr382/383, a measure of PTEN inactivation (Vazquez *et al*, 2000), was also normal in $Pten^{\Delta C/\Delta C}$ mice (Fig EV1D–F). Phosphorylation levels of AKT and mTOR in the downstream of PTEN were also normal (Fig EV1G). These results indicate that the deletion of the C-terminal 5-aa residues of PTEN minimally affects the stability and activity of the PTEN protein.

$Pten^{\Delta C/\Delta C}$ mice displayed normal levels of synaptic receptor proteins (NMDA, AMPA, and metabotropic glutamate receptors; Fig EV1H). Total/phosphorylation levels of synaptic plasticity-related proteins, including GSK3β, ERK, and p38, were also normal (Fig EV1I).

## Reduced excitatory synapse density in the $Pten^{\Delta C/\Delta C}$ hippocampus

To test whether the deletion of PTEN-PB has any influence on the density or function of neuronal synapses, we measured miniature excitatory postsynaptic currents (mEPSCs) in CA1 pyramidal neurons and dentate gyrus (DG) granule cells of the hippocampus, two brain regions with PTEN expression. Neurons in the CA1 region of postnatal day 17–21 (P17–21) $Pten^{\Delta C/\Delta C}$ mice displayed ~ 50% reduction in the frequency, but not amplitude, of mEPSCs (Fig EV2A). Intriguingly, these neurons displayed an increased frequency, but normal amplitude, of miniature inhibitory postsynaptic currents (mIPSCs; Fig EV2B).

Hippocampal DG granule neurons from $Pten^{\Delta C/\Delta C}$ mice (P18–23) displayed similar changes in mEPSCs (decreased frequency) and mIPSCs (increased frequency; Fig EV2C and D). These results indicate that PTEN C-terminal deletion leads to opposing changes in the frequency of mEPSCs and mIPSCs in both CA1 and DG regions, effects that would decrease the synaptic excitation/inhibition ratio, and thus the activity, of these neurons.

To determine whether network activity can compensate for these synaptic changes, we next measured spontaneous EPSCs and IPSCS (sEPSCs and sIPSCs) in the absence of tetrodotoxin, which blocks action potentials. CA1 neurons from $Pten^{\Delta C/\Delta C}$ mice (P17–21) displayed reduced sEPSC frequency (but not amplitude), but normal sIPSCs (Fig EV2E and F). This suggests that network activity can normalize the increased inhibitory transmission, but not the decreased excitatory transmission.

We further analyzed the excitatory synapse phenotype of $Pten^{\Delta C/\Delta C}$ mice (P21) by electron microscopic (EM) analysis. The density of the PSD (postsynaptic density) apposed to presynaptic axon terminals was significantly reduced in both CA1 and DG regions of the $Pten^{\Delta C/\Delta C}$ hippocampus relative to that of WT mice (Fig EV2G and H). In contrast, there was no difference in the length, width, or perforation (a measure of maturation) of the PSD in $Pten^{\Delta C/\Delta C}$ mice. These results collectively suggest that C-terminal PTEN deletion in mice leads to a reduction in the density of excitatory synapses in both CA1 and DG regions of the hippocampus that cannot be normalized by network activity.

## Suppressed NMDAR-mediated synaptic transmission and synaptic plasticity in the *Pten^{ΔC/ΔC}* hippocampus

Previous studies have shown that the PTEN-PB motif is required for synaptic translocation of PTEN during LTD and amyloid-β peptide (Aβ)-induced synaptic weakening (Jurado *et al*, 2010; Knafo *et al*, 2016). However, it remains unclear whether the PTEN-PB motif is also required for the regulation of other forms of synaptic plasticity.

To test this, we first measured basal excitatory transmission at Schaffer collateral-CA1 pyramidal (SC-CA1) synapses in the hippocampus of 4- to 5-week-old mice. *Pten^{ΔC/ΔC}* SC-CA1 synapses showed normal levels of basal excitatory synaptic transmission, as shown by input–output curve of fEPSP amplitudes plotted against fiber volley amplitudes (the strength of action potentials arriving at nerve terminals; Fig 1A), suggesting that excitatory synapses that survived the decrease in excitatory synapse number display normal synaptic transmission, in line with the normal mEPSC amplitude and PSD morphology (Fig EV2). In addition, paired-pulse facilitation at these synapses was normal (Fig 1B), suggestive of a normal probability of presynaptic release.

Induction of LTP at SC-CA1 synapses by high-frequency stimulation (HFS) was suppressed at *Pten^{ΔC/ΔC}* synapses compared with WT synapses (Fig 1C). The suppression was greater in the late maintenance phase of HFS-LTP than the early induction phase. In contrast, LTP induced by theta-burst stimulation (TBS) was normal at *Pten^{ΔC/ΔC}* SC-CA1 synapses (Fig 1D), consistent with previous results (Knafo *et al*, 2016).

Long-term depression induced by low-frequency stimulation (LFS) was suppressed at SC-CA1 synapses of *Pten^{ΔC/ΔC}* mice (P17–21; Fig 1E), consistent with a previous report (Knafo *et al*, 2016). In contrast, metabotropic glutamate receptor (mGluR)-dependent LTD was normal at SC-CA1 synapses of *Pten^{ΔC/ΔC}* mice (P17–21; Fig 1F). These results collectively suggest that PTEN-PB deletion in mice impacts specific forms of synaptic plasticity, suppressing HFS-LTP and LFS-LTD, but not TBS-LTP or mGluR-LTD.

High-frequency stimulation-LTP and LFS-LTD are known to require activation of NMDARs (Malenka & Bear, 2004), although TBS-LTP requires non-NMDAR components such as presynaptic mechanisms (Larson & Munkacsy, 2015). We thus directly measured NMDAR-mediated synaptic transmission at SC-CA1 synapses. The ratio of NMDAR- to AMPAR-mediated EPSCs (NMDA/AMPA ratio) was reduced at SC-CA1 synapses of *Pten^{ΔC/ΔC}* mice compared with WT synapses (Fig 1G). In addition, the amplitude but not frequency of NMDAR-mediated miniature excitatory postsynaptic currents (NMDA-mEPSCs) was decreased at *Pten^{ΔC/ΔC}* CA1 pyramidal neurons (Fig 1H). This, together with the abovementioned normal basal excitatory transmission mediated by AMPARs, suggests that NMDAR-mediated synaptic transmission is selectively decreased at *Pten^{ΔC/ΔC}* SC-CA1 synapses.

Measurements of the NMDA/AMPA ratio in layer II/III pyramidal neurons in the prelimbic region of the medial prefrontal cortex (mPFC) revealed that *Pten^{ΔC/ΔC}* synapses displayed a similar decrease in NMDA/AMPA ratio (Fig 1I). In addition, *Pten^{ΔC/ΔC}* NMDA currents decayed faster, a difference that could be attributable to several factors, including a decrease in the GluN2B component (Monyer *et al*, 1994; Vicini *et al*, 1998) or a decrease in the concentration of extracellular glycine (Chen *et al*, 2003), although

the immunoblot results indicate unaltered amounts of GluN2B subunit in the *Pten^{ΔC/ΔC}* brain (Fig EV1H).

These results suggest that NMDAR function is suppressed in both the hippocampus and mPFC of *Pten^{ΔC/ΔC}* mice, and may indicate that the reduction in HFS-LTP and LFS-LTD in the hippocampus is attributable to reduced NMDAR-dependent currents. In addition, given that there were no changes in the phosphorylation levels of synaptic plasticity-related proteins such as GluA1 (Ser-831 and Ser-845), ERK1/2, p38, and GSK3β (Fig EV1I), known to regulate synaptic plasticity (Roche *et al*, 1996; Mammen *et al*, 1997; Lee *et al*, 1998; Hayashi *et al*, 2000; Lee *et al*, 2000; Shi *et al*, 2001; Peineau *et al*, 2007), altered synaptic signaling at *Pten^{ΔC/ΔC}* hippocampal synapses is less likely to underlie the reduced HFS-LTP and LFS-LTD.

## Increased expression of the proline transporter SLC6A20 in the *Pten^{ΔC/ΔC}* brain

Previous studies have shown that PTEN can be localized to the nucleus and regulates diverse cellular processes, including nuclear signaling, gene expression, apoptosis, chromosome stability, and DNA repair (Gil *et al*, 2006; Gil *et al*, 2007; Planchon *et al*, 2008; Bassi *et al*, 2013). We thus reasoned that gene expression patterns might be altered in *Pten^{ΔC/ΔC}* mice.

To test this, we compared the transcriptomic profiles of WT and *Pten^{ΔC/ΔC}* brains (male) at P21, a developmental stage where most synaptic abnormalities were detected in our study. This RNA-Seq analysis revealed a total of nine differentially expressed genes (DEGs) between *Pten^{ΔC/ΔC}* and WT brains—three upregulated and six downregulated (Fig 2A). Eight of the nine DEGs were validated by quantitative reverse transcriptase-PCR (RT–qPCR; Fig EV1J). These DEGs encoded proteins with diverse functions, one of which was the *Slc6a20a* gene that encodes solute carrier family 6 member 20A (SLC6A20A; Fig 2A), a Na$^+$/Cl$^-$-dependent proline transporter, also known as XT3s1 and Xtrp3s1 (Smith *et al*, 1995; Nash *et al*, 1998; Kiss *et al*, 2002; Kowalczuk *et al*, 2005; Takanaga *et al*, 2005; Broer & Gether, 2012). Unlike the single human gene encoding SLC6A20, mice have two SLC6A20-related mouse genes, *Slc6a20a* (also known as XT3s1 and Xtrp3s1) and *Slc6a20b* (also known as XT3, Xtrp3, and SIT1) (Smith *et al*, 1995; Nash *et al*, 1998; Kiss *et al*, 2002; Kowalczuk *et al*, 2005; Takanaga *et al*, 2005; Broer & Gether, 2012), which are ~ 93 % identical in DNA sequences and ~ 91 % in aa sequences and show distinct tissue expression patterns; *Slc6a20a* is mainly expressed in the brain, kidney, small intestine, thymus, spleen, and lung, while *Slc6a20b* is only expressed in kidney and lung (Kowalczuk *et al*, 2005).

We next analyzed RNA-Seq data using gene set enrichment analysis (GSEA; software.broadinstitute.org/gsea/) (Subramanian *et al*, 2005). GSEA is a test that determines the enrichment of a list of genes (usually entire genes) ranked by expression for precurated gene sets, yielding an unbiased search of altered molecular, cellular, or pathway functions. A GSEA of *Pten^{ΔC/ΔC}* and WT transcriptomes using the gene sets in the C5 (gene ontology) category revealed two strongly enriched gene ontology functions: ribosome biogenesis and transmembrane transport, as shown by the list of the six most strongly enriched gene sets in the categories of cellular component, molecular function, and biological process (Fig 2B).

Enrichment for ribosome-related gene sets was negative (Fig 2B), meaning that downregulated genes mainly contributed to the enrichment. In contrast, enrichment for transmembrane transport-related gene sets was positive (Datasets EV1 and EV2; see "molecular function" and "biological process" sections), indicating that upregulated genes contributed strongly to the enrichment. Importantly, this positive enrichment is in line with the increased expression of *Slc6a20a*. These GSEA results, together with the DEG results, suggest that deletion of the PTEN-PB motif induces an increase in transmembrane transport-related functions.

Given that the *Slc6a20a* gene showed an increase in the expression in our RNA-Seq analysis, we further analyzed the expression of *Slc6a20a* at the protein level using mouse brain tissues. Using immunoblotting, we first found that the levels of the SLC6A20 protein were increased in the hippocampus of $Pten^{\Delta C/\Delta C}$ mice (P21; Fig 2C). This protein band likely represents the SLC6A20A protein because *Slc6a20a* (not *Slc6a20b*) is the major gene expressed in the brain, as mentioned above.

We next analyzed specific brain regions and cell types that express *Slc6a20* mRNA using fluorescence *in situ* hybridization (FISH) and a *Slc6a20* probe that detects both *Slc6a20a* and *Slc6a20b*. *Slc6a20* mRNA was widespread in various brain regions, including the meninges, choroid plexus, cortex, hippocampus, and thalamus (Figs EV3–EV5), similar to previous results (Kowalczuk *et al*, 2005; Takanaga *et al*, 2005; Dahlin *et al*, 2009). Notably, the meningeal and choroid plexus localization of *Slc6a20* mRNAs agreed with the reported enrichment of *Slc6a20a* expression in pericytes around capillaries, known to regulate BBB functions (Hall *et al*, 2014; He

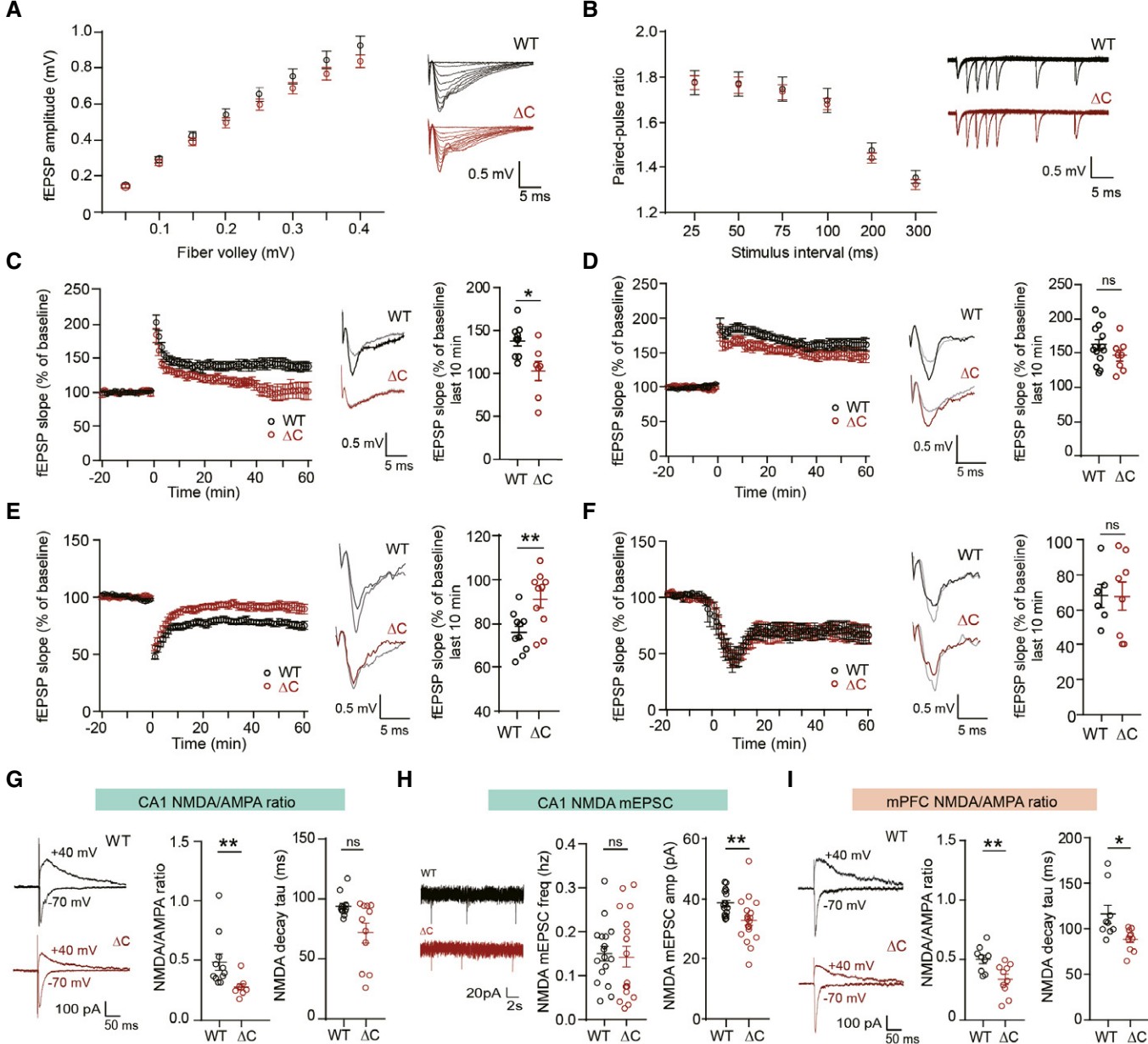

**Figure 1.**

**Figure 1.  Suppressed NMDAR-mediated synaptic transmission and plasticity in the *Pten^{ΔC/ΔC}* hippocampus.**

A    Normal basal excitatory synaptic transmission at SC-CA1 synapses of *Pten^{ΔC/ΔC}* mice (4–5 weeks), indicated by the input–output curve of the amplitudes of field excitatory postsynaptic potentials (fEPSPs) plotted against stimulus intensities. ($n$ = 10 slices from four mice for WT and nine (4) for ΔC, genotype $P$ = 0.8378, two-way repeated measures ANOVA with Bonferroni's multiple comparison test). The error bars represent Standard error of the mean SEM (see Appendix Table S1 "Statistics" for full details.

B    Normal paired-pulse facilitation at SC-CA1 synapses of *Pten^{ΔC/ΔC}* mice (4–5 weeks), indicated by fEPSP slope ratios plotted against inter-stimulus intervals. ($n$ = 11 slices from four mice for WT and 11 (4) for ΔC, genotype $P$ = 0.7376, two-way repeated measures ANOVA with Bonferroni's test). The error bars represent SEM.

C, D  Suppressed HFS-LTP and normal TBS-LTP at SC-CA1 synapses of *Pten^{ΔC/ΔC}* mice (4–6 weeks). The gray traces represent baseline fEPSP prior to LTP induction. The error bars represent SEM. ($n$ = 9 slices from six mice for WT and seven (6) for ΔC for HFS-LTP, and 14 (9) for WT and eight (5) for ΔC for TBS-LTP, *$P$ < 0.05, ns, not significant, Mann–Whitney *U*-test and Student's *t*-test.

E    Suppressed LFS-LTD at SC-CA1 synapses of *Pten^{ΔC/ΔC}* mice (P17–21). ($n$ = 10 slices from six mice for WT and ΔC, **$P$ < 0.01, Student's *t*-test). The gray traces represent baseline fEPSP prior to LTD induction. The error bars represent SEM.

F    Normal mGluR-LTD at SC-CA1 synapses of *Pten^{ΔC/ΔC}* mice (P17–21). ($n$ = 6 slices from five mice for WT and eight (5) for ΔC, ns, not significant, Mann–Whitney *U*-test). The gray traces represent baseline fEPSP prior to LTD induction. The error bars represent SEM.

G    Reduced NMDA/AMPA ratio at SC-CA1 synapses of *Pten^{ΔC/ΔC}* mice (P17–21). The NMDAR component was obtained 60 ms after stimulation. (ratio, $n$ = 11 cells from eight mice for WT and eight (8) for ΔC; decay, $n$ = 12 cells from eight mice for WT and 11 (8) for ΔC, **$P$ < 0.01, ns, not significant, Mann–Whitney *U*-test). The error bars represent SEM.

H    Reduced amplitude but not frequency of NMDA-mEPSCs at *Pten^{ΔC/ΔC}* CA1 pyramidal neurons. ($n$ = 17 cells from four mice for WT and 18 (4) for ΔC, **$P$ < 0.01, ns, not significant, Mann–Whitney *U*-test, Student's *t*-test). The error bars represent SEM.

I    Reduced NMDA/AMPA ratio at synapses in layer II/III pyramidal neurons in the prelimbic region of the mPFC of *Pten^{ΔC/ΔC}* mice (P20–21). The NMDAR component was obtained 60 ms after stimulation. ($n$ = 9 cells from three mice for WT and 10 (3) for ΔC, *$P$ < 0.05, **$P$ < 0.01, Student's *t*-test, Mann–Whitney *U*-test). The error bars represent SEM.

Data information: (C–F) Results from the last 10-min recordings are shown in scatter plots.

*et al*, 2016; Vanlandewijck *et al*, 2018). SLC6A20A protein expression, revealed by X-gal staining of SLC6A20A protein fused with β-galactosidase in *Slc6a20a^{+/−}* mice (see below for further details), was also detected in similar brain regions (Fig EV3B).

*Slc6a20* mRNA was detected in distinct cell types, exhibiting high expression in astrocytes (GFAP-positive) and microglia (Aif1-positive), modest expression in glutamatergic (Vglut1/2-positive) neurons, and little expression in GABAergic (Gad1/2-positive) neurons (Fig EV4 and EV5). In addition, most *Slc6a20*-expressing cells colocalized with *Pten*-expressing cells (Appendix Fig S1), likely owing to the more widespread expression of *Pten*, which suggests the possibility of cell-autonomous interplay between PTEN and SLC6A20. These results collectively suggest that SLC6A20A expression is increased in the *Pten^{ΔC/ΔC}* brain.

### Increased nuclear localization of β-catenin in the *Pten^{ΔC/ΔC}* brain

How might the C-terminal deletion in PTEN induce changes in gene expression? A previous study has shown that a PB (PDZ-binding)-lacking mutant PTEN protein can destabilize adherens junctions in retinal epithelial cells and cause the nuclear translocation of β-catenin and altered TCF/LEF-dependent gene expression (Kim *et al*, 2008). In the brain, β-catenin links N-cadherins with the actin cytoskeleton at excitatory synapses and critically regulates the morphogenesis of dendritic spines (Yu & Malenka, 2003). In addition, excitatory synaptic localization of β-catenin is promoted by neuronal activation through tyrosine phosphorylation of β-catenin at Tyr-654 (Tai *et al*, 2007). More recently, β-catenin has been shown to associate with synaptic proteins including Shank3 and be targeted to the nucleus to regulate HDAC2 expression (Qin *et al*, 2018).

Given that PTEN-PB deletion leads to a decrease in the number of dendritic spines, we reasoned that β-catenin might display altered subcellular localization in *Pten^{ΔC/ΔC}* mice. Previous studies have shown that phosphorylation of β-catenin at two serine residues

(Ser-552 and Ser-675) by AKT and PKA, respectively, promotes the stability of the protein by inhibiting ubiquitination and the nuclear localization and transcriptional activity of β-catenin through, i.e., enhanced association with CREB-binding protein in the case of Ser-675 (Hino *et al*, 2005; Taurin *et al*, 2006; Fang *et al*, 2007).

Intriguingly, there was a significant increase in the phosphorylation of β-catenin at Ser-675 but not at Ser-552 in the nucleus-enriched P1 fraction of the *Pten^{ΔC/ΔC}* brain, without a change in total β-catenin levels (Fig 2D). In addition, whole-brain lysates (not P1 fraction) of the *Pten^{ΔC/ΔC}* brain displayed normal levels of β-catenin phosphorylation at both Ser-675 and Ser-552 (Fig 2E).

In addition, when a chromatin immunoprecipitation (ChIP) analysis was performed to test whether β-catenin binds to the promoter region of the *Slc6a20a* gene, β-catenin bound more tightly to the *Slc6a20a* promoter in the *Pten^{ΔC/ΔC}* brain, compared with WT brain (Fig 2F). These results suggest that PTEN-PB deletion promotes the nuclear localization of β-catenin and β-catenin-dependent transcription of *Slc6a20a*.

### Decreased extracellular proline and glycine levels in the *Pten^{ΔC/ΔC}* brain

Previous studies have shown that mouse, rat, and human SLC6A20 proteins mediate proline transport (Kowalczuk *et al*, 2005; Takanaga *et al*, 2005; Broer *et al*, 2009). We thus measured whether the brain levels of proline was changed in the whole brain of *Pten^{ΔC/ΔC}* mice by enzyme-linked immunosorbent assay (ELISA) but found no genotype difference between WT and *Pten^{ΔC/ΔC}* brains (Fig 3A). However, when microdialysis was used to measure extracellular proline levels, there was a decrease in proline levels in the *Pten^{ΔC/ΔC}* mice, compared with WT mice (Fig 3B and C) suggesting that increased levels of SLC6A20A in the brain induce a decrease in extracellular but not total levels of proline.

Although previous studies have shown mouse SLC6A20A minimally mediates glycine transports (Kowalczuk *et al*, 2005), we noted

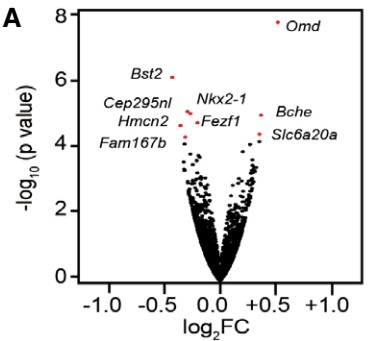

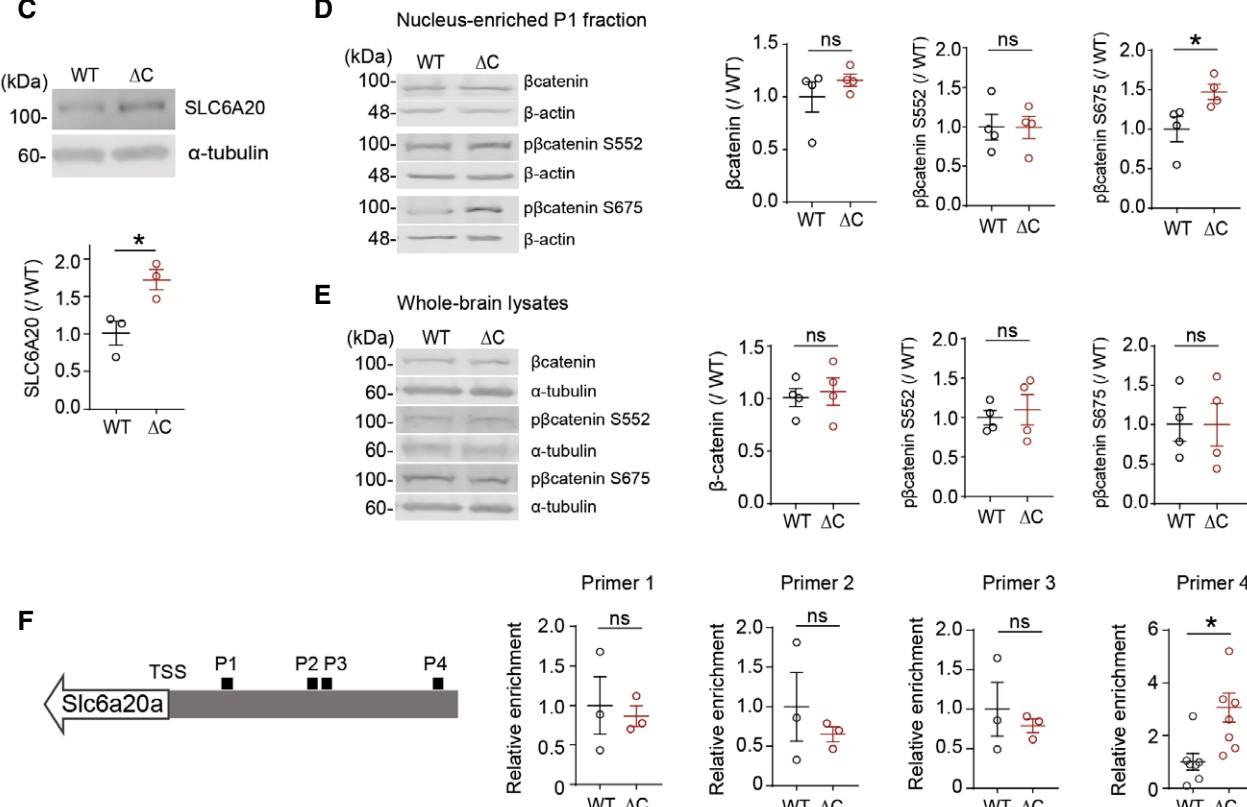

**Figure 2.**

**Figure 2. Increased expression of the *Slc6a20a* glycine transporter gene and decreased brain glycine levels in the *Pten*^ΔC/ΔC^ mice.**

A A volcano plot showing DEGs (adjusted *P* value < 0.1) derived from RNA-Seq results for *Pten*^ΔC/ΔC^ and WT mice (P21). For the calculation of adjusted *P* value, we have employed the R package DESeq2 where the *P*-values obtained by the Wald test are corrected for multiple testing using the Benjamini and Hochberg method. (*n* = 3 mice for WT and ΔC). See also Dataset EV1 for the full RNA-Seq results.

B GSEA analysis of RNA-Seq results from *Pten*^ΔC/ΔC^ and WT mice (P21). Transcripts ranked by levels of expression (ΔC/WT ratio) were tested for enrichment in precurated gene sets in the C5 (gene ontology) category. NES, normalized enrichment score; positive and negative enrichments indicate a greater contribution of up- and downregulated genes, respectively, to the enrichments (see also Dataset EV2 for the full GSEA results).

C Immunoblot validation of the increase in SLC6A20 protein levels in hippocampal lysates of *Pten*^ΔC/ΔC^ mice (P21). We denoted the antibody as "SLC6A20" here, not "SLC6A20A", because this antibody was generated using a synthetic peptide sequence (YNEPSNNCQKHAI) that is commonly present in SLC6A20A and SLC6A20B, being a pan-SLC6A20 antibody (Thermo Fisher). (*n* = 3 mice for WT and ^Δ^C, *\*P* < 0.05, Student's *t*-test). The error bars represent SEM.

D, E Increased phosphorylation of β-catenin at Ser-675 but not at Ser-552 in the nucleus-enriched P1 fraction but not in whole-brain lysates of *Pten*^ΔC/ΔC^ mice (3 months). Note that total levels of β-catenin were not changed in the P1 fraction or whole-brain lysates. (*n* = 4 mice for WT-WB/P1 and ΔC-WB/P1, *\*P* < 0.05, ns, not significant, Student's *t*-test). The error bars represent SEM.

F Increased binding of β-catenin in the promoter region of the *Slc6a20a* gene in *Pten*^ΔC/ΔC^ mice (P21), revealed by chromatin immunoprecipitation (ChIP) assay. Note that β-catenin binding was increased selectively in the target region for the primer set #4 indicated in the diagram. TSS, transcriptional start site; P1–4; promoter regions targeted by the four primer sets. (*n* = 7 mice for WT-P4 and ΔC-P4 and 3 mice for WT-P1/2/3 and ΔC-P1/2/3, *\*P* < 0.05, ns, not significant, Student's *t*-test). The error bars represent SEM.

Source data are available online for this figure.

that *Pten*^ΔC/ΔC^ mice show decreased NMDAR currents and that glycine, whose extracellular concentrations can be regulated by glycine transporters (Harvey & Yee, 2013), acts as a co-agonist of NMDARs. We thus tested whether glycine levels were changed in the brain of *Pten*^ΔC/ΔC^ mice and found that both whole-brain and extracellular glycine levels were decreased (Fig 3A–C). These results suggest that increased levels of SLC6A20A proteins in the *Pten*^ΔC/ΔC^ brain may lead to decreased whole-brain and extracellular glycine levels through enhanced glycine transport into brain-resident cells or outside the brain.

### Enhanced repetitive climbing in *Pten*^ΔC/ΔC^ mice

To determine whether the synaptic and molecular abnormalities observed in *Pten*^ΔC/ΔC^ mice are associated with behavioral alterations, we subjected *Pten*^ΔC/ΔC^ mice (2–4 months) to a battery of behavioral tests. *Pten*^ΔC/ΔC^ mice displayed largely normal levels of social interaction but mildly impaired social novelty recognition in the three-chamber test compared with WT mice (Appendix Fig S2A and B). In addition, these mice displayed normal repetitive behaviors, such as self-grooming, digging, and marble burying (Appendix Fig S2C–E).

However, continuous monitoring of mouse behaviors for ~ 72 h in Laboras cages revealed enhanced climbing, a type of repetitive behavior in rodents characterized by over-hanging in the wire cage lid (Protais *et al*, 1976; Riffee *et al*, 1979; Wilcox *et al*, 1979; Cabib & Puglisi-Allegra, 1985), in *Pten*^ΔC/ΔC^ mice at both juvenile and adult stages; in contrast, these mice displayed normal self-grooming, another form of repetitive behavior, at both stages in Laboras cages (Fig 3D; Appendix Fig S2F and G).

*Pten*^ΔC/ΔC^ mice displayed normal levels of locomotor activity in open field and Laboras tests, lack of anxiety-like behavior in elevated plus-maze and light-dark tests, and intact learning and memory in novel object recognition and Morris water maze tests (Appendix Fig S3). These mice also showed normal levels of other behaviors: long-term (7-day) spatial memory in the Morris water maze, nesting score, mother seeking after brief separation, and juvenile play (Appendix Fig S4). These results collectively suggest that *Pten*^ΔC/ΔC^ mice display a selective increase in repetitive climbing behavior.

### NMDAR activation by D-cycloserine normalizes synaptic plasticity and repetitive climbing in *Pten*^ΔC/ΔC^ mice

The suppressed NMDAR function at *Pten*^ΔC/ΔC^ synapses associated with reduced brain glycine levels might be normalized using agonists that act on the glycine-binding site of NMDARs (Fig 4A). To test this, we used D-cycloserine (DCS), a glycine-site agonist of NMDARs. Continuous exposure of hippocampal slices to DCS (20 μM) during recordings significantly increased NMDAR-dependent HFS-LTP at SC-CA1 synapses in hippocampal slices from *Pten*^ΔC/ΔC^ mice (4–5 weeks) compared with vehicle treatment (Fig 4B). In contrast, DCS had no effect on HFS-LTP at WT synapses. DCS treatment (10 μM) also normalized LFS-LTD at SC-CA1 synapses of *Pten*^ΔC/ΔC^ mice (P17–22) without affecting WT synapses (Fig 4C).

Behaviorally, acute DCS treatment (20 mg/kg) significantly decreased excessive climbing in *Pten*^ΔC/ΔC^ mice (2–5 months; Fig 4D). These results suggest that suppressed NMDAR function induces excessive climbing and that normalizing NMDR function by DCS treatment rescues this effect.

### NMDAR activation by GlyT1 inhibition normalizes synaptic plasticity and repetitive climbing in *Pten*^ΔC/ΔC^ mice

In addition to directly activating NMDARs with DCS, we tested the effects of increasing brain glycine levels using sarcosine, a competitive GlyT1-specific antagonist that can increase glycine levels around NMDARs (Harvey & Yee, 2013) likely through the inhibition of glycine uptake by GlyT1 or glycine release through heteroexchange of sarcosine for glycine by GlyT1 (Herdon *et al*, 2001) (Fig 4E). Sarcosine treatment normalized HFS-LTP at SC-CA1 synapses of *Pten*^ΔC/ΔC^ mice (4–5 weeks), without affecting WT synapses (Fig 4F). In addition, sarcosine normalized LFS-LTD at *Pten*^ΔC/ΔC^ SC-CA1 synapses (P16–21), without affecting WT synapses (Fig 4G).

Behaviorally, sarcosine rescued excessive climbing in *Pten*^ΔC/ΔC^ mice (2–5 months), but had no effect on WT mice (Fig 4H). These results collectively suggest that NMDAR activation by blocking GlyT1 normalizes NMDAR-dependent synaptic plasticity and excessive climbing behavior, effects similar to those of DCS.

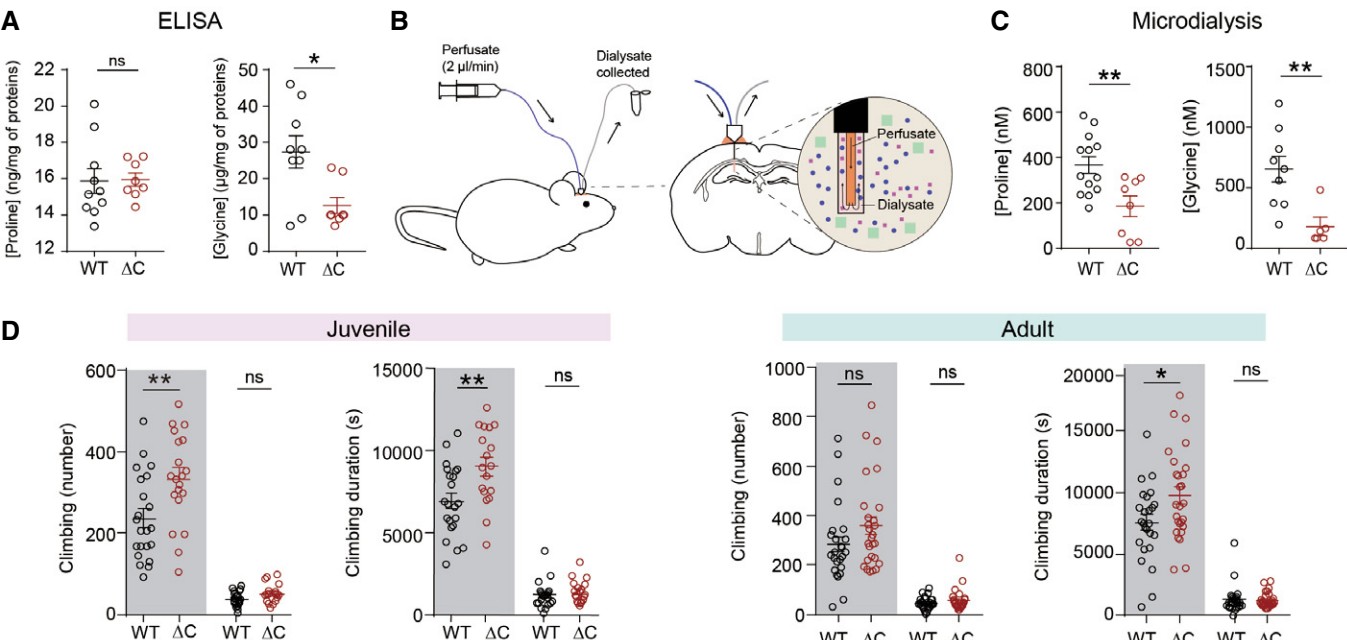

**Figure 3. Decreased extracellular proline and glycine levels in the *Pten^{ΔC/ΔC}* brain and increased climbing in juvenile and adult *Pten^{ΔC/ΔC}* mice.**

A  Normal proline but decreased whole-brain glycine levels in the brains of *Pten^{ΔC/ΔC}* mice (P21), as determined by ELISA analysis of the S1 fraction, obtained after the removal of unbroken cells and large debris (see Materials and Methods for details). (proline, $n = 10$ mice for WT and 8 for ΔC, glycine, $n = 9$ mice for WT and eight for ΔC, *$P < 0.05$, ns, not significant, Student's *t*-test. The error bars represent SEM.

B, C  Decreased extracellular levels of proline and glycine in the brains of *Pten^{ΔC/ΔC}* mice (2–4 month), as determined by microdialysis in the hippocampus (see Materials and Methods for details). (proline, $n = 13$ mice for WT and 8 for ΔC, **$P < 0.01$, Student's *t*-test; glycine, $n = 9$ for WT and 5 for ΔC, **$P < 0.01$, Student's *t*-test, Mann–Whitney *U*-test). The error bars represent SEM.

D  Increased repetitive climbing in juvenile and adult *Pten^{ΔC/ΔC}* mice at P30 (juvenile) and 2–4 months (adult), as indicated by frequency and time spent climbing in Laboras cages, where mouse movements were continuously monitored for 72 h. Shaded and unshaded periods; 12-h light-off and light-on periods over 72 h. ($n = 22$ mice for WT and 18 mice for ΔC for P30, and 25 for WT and 27 for ΔC for 2–4 months, *$P < 0.05$, **$P < 0.01$, ns, not significant, Mann–Whitney *U*-test, Student's *t*-test). The error bars represent SEM.

## Antisense knockdown of *Slc6a20a* expression increases whole-brain glycine levels and NMDAR function and rescues repetitive climbing in *Pten^{ΔC/ΔC}* mice

If the abnormally increased expression of *Slc6a20a* is important for the reduced NMDAR function and repetitive climbing observed in *Pten^{ΔC/ΔC}* mice, normalizing *Slc6a20a* expression in these mice should rescue NMDAR function and repetitive climbing. To this end, we knocked down *Slc6a20a* expression using an *Slc6a20a*- (but not *Slc6a20b*-) specific antisense oligonucleotide (ASO) method that involves the use of modified antisense oligonucleotides with enhanced cell permeability, an approach that has recently proven to be useful in a clinical setting (Rinaldi & Wood, 2018). An *Slc6a20a*-ASO was generated based on the previously published antisense sequence for *Slc6a20a* (Anas *et al*, 2008) and unilaterally injected into ventricular spaces of target mice (Fig 5A). RT–qPCR and immunoblot analyses after the injection confirmed the decreased levels of *Slc6a20a* mRNAs and SLC6A20 proteins in the brain of *Slc6a20a*-ASO-injected WT mice relative to saline-injected WT mice (Fig 5B and C), suggesting that *Slc6a20a*-ASO decreases SLC6A20A protein levels mainly through mRNA degradation. *Slc6a20a*-ASO induced a greater decrease in *Slc6a20a* mRNAs in *Pten^{ΔC/ΔC}* mice.

When injected into the brain of *Pten^{ΔC/ΔC}* mice, *Slc6a20a*-ASO partially normalized whole-brain glycine levels, as supported by the

lack of a significant difference between saline-treated WT mice and ASO-treated *Pten^{ΔC/ΔC}* mice, although there was no significant difference between saline- and ASO-treated *Pten^{ΔC/ΔC}* mice (Fig 5D), suggesting that the reduced whole-brain glycine levels in *Pten^{ΔC/ΔC}* mice partially involves enhanced *Slc6a20a* expression. This increase in glycine levels was associated with increased NMDAR currents, as shown by the comparable levels of NMDA/AMPA ratios in ASO-treated WT and *Pten^{ΔC/ΔC}* mice (Fig 5E). In contrast, the mEPSC frequency, which is decreased in naive *Pten^{ΔC/ΔC}* mice, was unaffected by *Slc6a20a*-ASO (Fig 5F).

Importantly, behavioral tests showed that repetitive climbing in *Pten^{ΔC/ΔC}* mice was fully normalized to a level comparable to that in WT mice, whereas WT mice were not affected by *Slc6a20a*-ASO (Fig 5G). Unlike climbing, self-grooming behavior was not affected by *Slc6a20a*-ASO in *Pten^{ΔC/ΔC}* mice (Fig 5H). Collectively, these results indicate that *Slc6a20a*-ASO selectively rescues climbing behavior, and suggest that increased SLC6A20A expression and consequent reductions in glycine levels and NMDAR function contribute to the repetitive climbing in *Pten^{ΔC/ΔC}* mice.

### *Slc6a20a*-mutant mice show abnormal increases in extracellular glycine levels and NMDAR function

Because the *in vivo* functions of SLC6A20A in mice have not been investigated, and our data implicate SLC6A20A in the regulation of

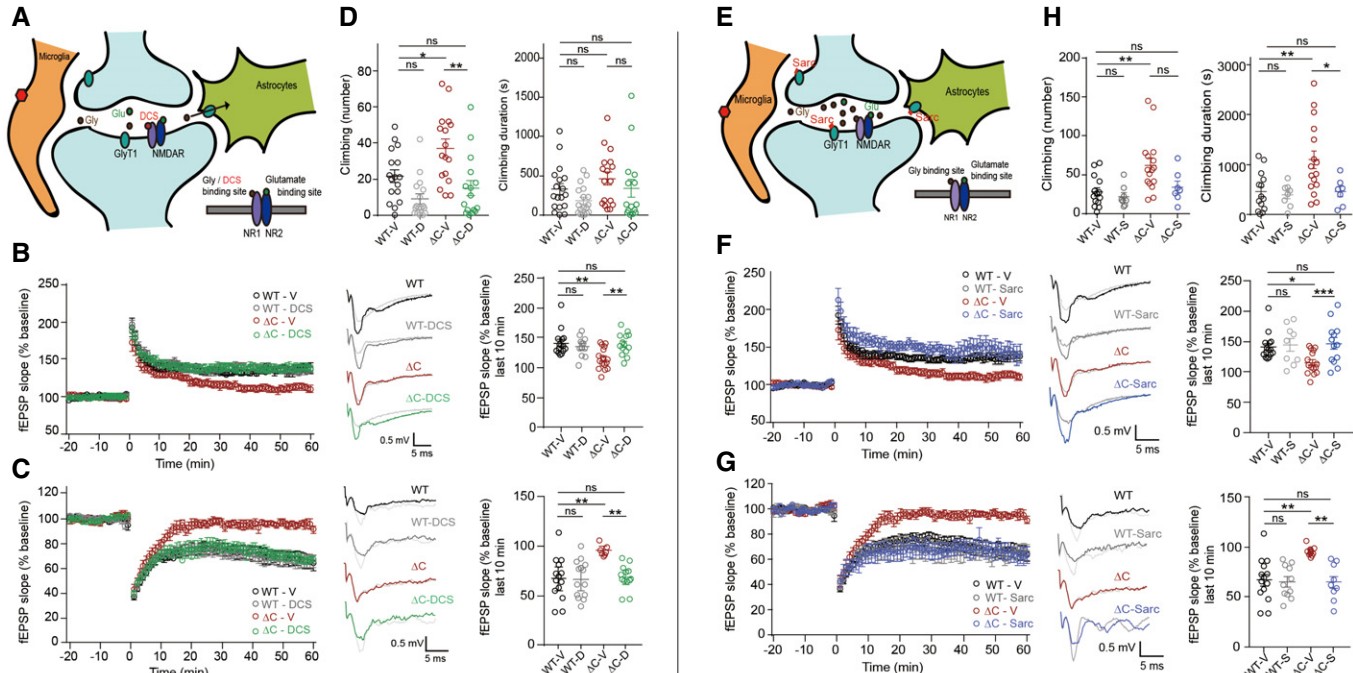

**Figure 4. DCS-dependent NMDAR activation and sarcosine-dependent GlyT1 antagonization normalize synaptic plasticity and repetitive climbing in *Pten*^ΔC/ΔC^ mice.**

A   Schematic showing that DCS directly binds to the GluN1 subunit of NMDARs and activates NMDARs. Note that enhanced expression of SLC6A20 in *Pten*^ΔC/ΔC^ microglia and astrocytes surrounding the indicated synapse may decrease synaptic levels of glycine and suppress NMDAR function.

B   Normalization of NMDAR-dependent HFS-LTP at synapses of *Pten*^ΔC/ΔC^ mice (4–5 weeks) by DCS treatment (20 μM), as shown by fEPSP slopes. ($n = 15$ slices from nine mice for WT-V/vehicle, 11 (5) for WT-D/DCS, 16 (10) for ΔC-V, 13 (7) for ΔC-D, **$P < 0.01$, ns, not significant, two-way ANOVA with Tukey's test). The gray traces represent the baseline fEPSP prior to LTP induction. The error bars represent SEM.

C   Normalization of NMDAR-dependent LFS-LTD at synapses of *Pten*^ΔC/ΔC^ mice (P17–22) by DCS treatment (10 μM), as shown by fEPSP slopes. ($n = 14$ slices from six mice for WT-V, 13 (4) for WT-D, seven (4) for ΔC-V, 13 (3) for ΔC-D, **$P < 0.01$, ns, not significant, two-way ANOVA with Tukey's test). The gray traces represent baseline fEPSP prior to LTD induction. The error bars represent SEM.

D   Normalization of excessive climbing frequency (but not duration) by treatment of synapses in *Pten*^ΔC/ΔC^ mice (2–5 months) with DCS (20 mg/kg). ($n = 17$ mice for WT-V, 17 for WT-D, 16 for ΔC-V, 16 for ΔC-D, *$P < 0.05$, **$P < 0.01$, ns, not significant, two-way ANOVA with Tukey's test). The error bars represent SEM.

E   Schematic showing that NMDAR activation can be induced indirectly through sarcosine-dependent antagonization of GlyT1, a known glycine transporter, and resultant increases in glycine levels around the synapse.

F   Normalization of NMDAR-dependent HFS-LTP at synapses of *Pten*^ΔC/ΔC^ mice (4–5 weeks) by sarcosine treatment (750 μM), as shown by fEPSP slopes. Note that the data for WT-V and ΔC-V are identical to those shown in Fig 4B because the whole experiments were performed together; we generated independent figures for DCS and sarcosine results for the clear presentation of the data. ($n = 15$ slices from nine mice for WT-V/vehicle, nine (4) for WT-Sarc/Sarcosine, 16 (10) for ΔC-V/Vehicle, 18 (4) for ΔC-S, *$P < 0.05$, ***$P < 0.001$, ns, not significant, two-way ANOVA with Tukey's test). The gray traces represent the baseline fEPSP prior to LTP induction. The error bars represent SEM.

G   Normalization of NMDAR-dependent LFS-LTD at synapses of *Pten*^ΔC/ΔC^ mice (16–21 days) by sarcosine treatment (750 μM), as shown by fEPSP slopes. Note that the data for WT-V and ΔC-V are identical to those shown in Fig 4C because the whole experiments were performed together; we generated independent figures for DCS and sarcosine results for the clear presentation of the data. ($n = 14$ slices from six mice for WT-V, 11 (7) for WT-Sarc, seven (4) for ΔC-V, nine (6) for ΔC-Sarc, **$P < 0.01$, ns, not significant, two-way ANOVA with Tukey's multiple comparison). The gray traces represent baseline fEPSP prior to LTD induction. The error bars represent SEM.

H   Normalization of excessive climbing duration (but not frequency) by treatment of synapses of *Pten*^ΔC/ΔC^ mice (2–5 months) with sarcosine (100 mg/kg). ($n = 14$ mice for WT-V, eight for WT-S, 16 for ΔC-V, seven for ΔC-S, *$P < 0.05$, **$P < 0.01$, ns, not significant, two-way ANOVA with Tukey's test).

brain glycine homeostasis, we attempted to knock out *Slc6a20a* in mice. To this end, we generated a new mouse line that lacks exon 3 of the *Slc6a20a* gene (Fig 6A; Appendix Fig S5A and B). The Mendelian ratio of WT, heterozygous *Slc6a20a*^+/−^, and homozygous *Slc6a20a*^−/−^ mice was 1.00:1.93:0.55 (106 mice from 13 litters at P14), indicative of decreased survival in *Slc6a20a*^−/−^ mice. Gross morphologies of the brain in *Slc6a20a*^+/−^ and *Slc6a20a*^−/−^ mice were largely normal (Appendix Fig S5C). qRT–PCR analyses indicated that *Slc6a20a*, but not *Slc6a20b*, mRNA levels were decreased

in the brains of both *Slc6a20a*^+/−^ and *Slc6a20a*^−/−^ mice in a dose-dependent manner (Fig 6B).

Immunoblot analyses indicated that SLC6A20A protein levels in the brain of *Slc6a20a*^+/−^ and *Slc6a20a*^−/−^ mice were decreased by ~ 32 and ~ 77% (Fig 6C), demonstrating that this knockout strategy was successful. The reason for the protein levels of SLC6A20A not reaching a 100% decrease in *Slc6a20a*^−/−^ mice is unclear, although it could be that the remaining band represents the SLC6A20B (not SLC6A20A) protein that is also expressed in the

brain, as supported by the presence of detectable *Slc6a20b* mRNAs (Fig 6B), and can be recognized by the pan-SLC6A20 antibody (Fig 6C).

Importantly, microdialysis analyses indicated extracellular brain glycine levels were increased in both *Slc6a20a*[+/−] and *Slc6a20a*[−/−] mice (Fig 6D). In addition, extracellular proline levels were increased only in *Slc6a20a*[−/−] mice, although there was an increasing tendency in *Slc6a20a*[+/−] mice. Therefore, *Slc6a20a* deletion has greater impacts on extracellular levels of glycine than those of proline in the mouse brain.

Lastly, *Slc6a20a*[+/−] mice displayed a substantially increased NMDA/AMPA ratio in pyramidal neurons in the CA1 region of the hippocampus (Fig 6E) but normal AMPAR-mediated mEPSC frequency and amplitude (Fig 6F), suggestive of increased NMDAR function, similar to the previously reported enhanced NMDAR function in mice lacking GlyT1 (Tsai *et al*, 2004b; Gabernet *et al*, 2005).

These results collectively suggest that SLC6A20A regulates extracellular glycine and proline levels and NMDAR function in the mouse brain.

## Mouse and human SLC6A20 proteins mediate both proline and glycine transports, and human neurons express SLC6A20 proteins

The results mentioned thus far suggest that SLC6A20A proteins mediate the transport of glycine in addition to proline in the mouse brain. To obtain direct evidence supporting that SLC6A20A mediates glycine transport, we tested four different types of human and mouse SLC6A20 protein variants; two different splice variants of human SLC6A20 (V1 and V2) and two different mouse SLC6A20 proteins encoded by two independent Slc6a20 genes (*Slc6a20a* and *Slc6a20b*; type a is widely expressed including the brain, type b is mainly expressed in the kidney and lung) (Kowalczuk *et al*, 2005)

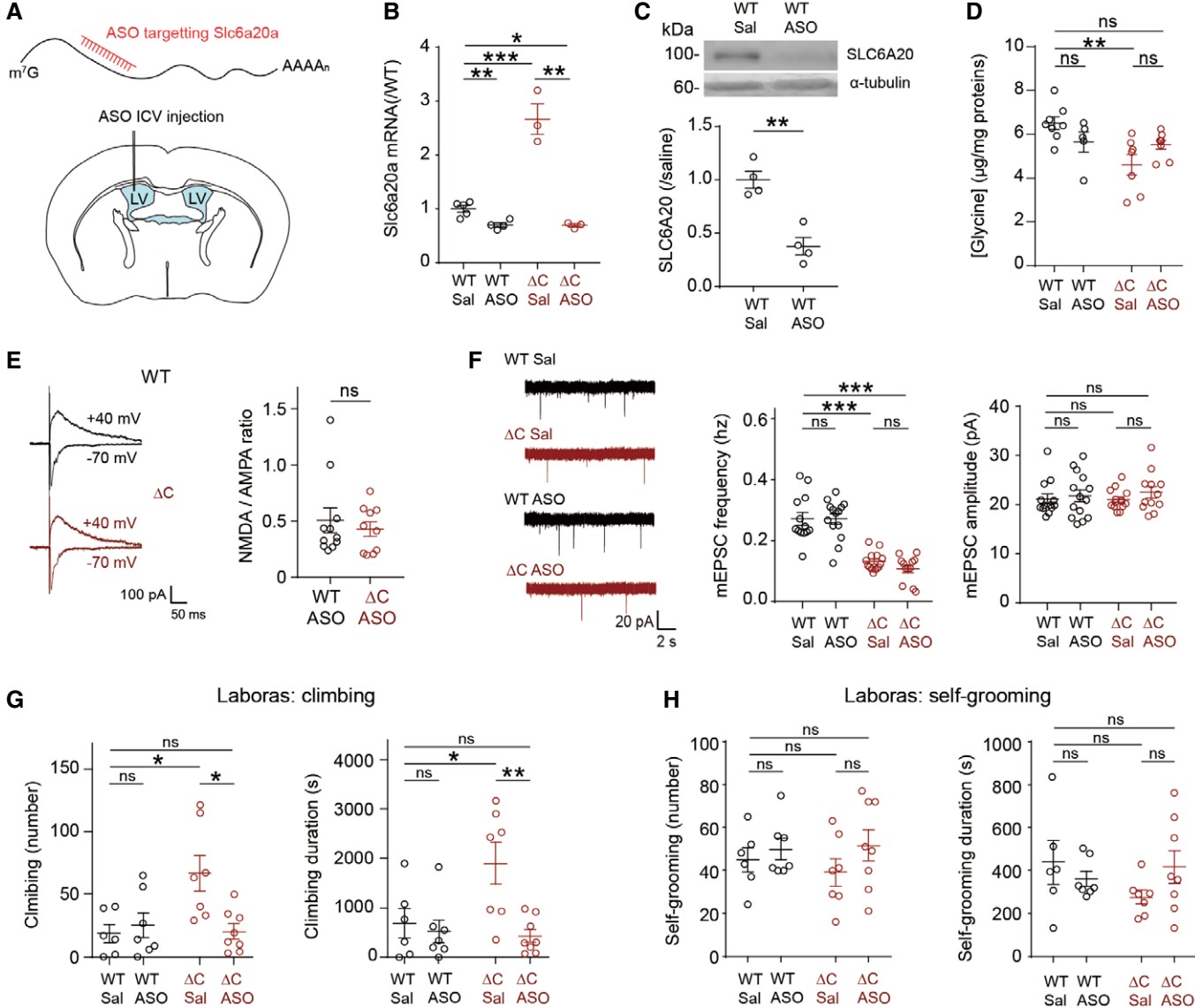

**Figure 5.**

**Figure 5.   Knockdown of SLC6A20A expression by antisense oligonucleotide increases whole-brain glycine levels and NMDAR function and rescues repetitive climbing in *Pten^{ΔC/ΔC}* mice.**

A   Schematic representation for the intracerebroventricular (ICV) injection of *Slc6a20a* antisense oligonucleotide (ASO) into the lateral ventricle (LV) for the reduction of *Slc6a20a* expression in the whole mouse brain.

B   *Slc6a20a*-ASO decreases the levels of *Slc6a20a* mRNAs in the brain of WT and *Pten^{ΔC/ΔC}* mice (P20–32), as shown by qRT–PCR analysis performed 1–2 weeks after injection. (*n* = 5 mice for WT-Sal/saline, four for WT-ASO, three for ΔC-Sal, and three for ΔC-ASO, \*$P < 0.05$, \*\*$P < 0.01$, \*\*\*$P < 0.001$, Student's *t*-test for each comparison). The error bars represent SEM.

C   *Slc6a20a*-ASO decreases the levels of SLC6A20 proteins in the brain of WT mice (2–4 months), as shown by immunoblot analysis of whole-brain lysates 6 days after injection. (*n* = 4 mice, \*\*$P < 0.01$, Student's *t*-test). The error bars represent SEM.

D   *Pten^{ΔC/ΔC}* mice (2–4 months) treated with *Slc6a20a*-ASO display partially normalized levels of whole-brain glycine, as shown by the lack of difference between saline-treated WT and ASO-treated *Pten^{ΔC/ΔC}* mice. Note that the glycine levels observed here are ~ 4 times lower than those measured in Fig 3A, which could be attributable to different mouse ages (P21 in Fig 3A vs. 2–4 months in Fig 5D), absence and presence of ASO injection, or lot-to-lot variation of the ELISA kits. (*n* = 8 mice for WT-saline, five for WT-ASO, seven for ΔC-saline, and eight for ΔC-ASO, \*\*$P < 0.01$, ns, not significant, two-way ANOVA with Tukey's test). The error bars represent SEM.

E   *Pten^{ΔC/ΔC}* mice (P20–37) treated with *Slc6a20a*-ASO display an NMDA/AMPA ratio at hippocampal SC-CA1 synapses that is comparable to that in ASO-treated WT mice. (*n* = 11 neurons from five mice for WT-ASO, 10 (4) for ΔC-ASO, ns, not significant, Mann–Whitney *U*-test). The error bars represent SEM.

F   *Pten^{ΔC/ΔC}* mice (P19–37) treated with *Slc6a20a*-ASO display unaltered mEPSC frequency and amplitude in CA1 pyramidal neurons, as compared with ASO-untreated *Pten^{ΔC/ΔC}* mice. Note that mEPSCs in WT mice are not affected by *Slc6a20a*-ASO treatment. (*n* = 13 neurons from three mice for WT-saline, 14 (2) for WT-ASO, 14 (3) for ΔC-saline, and 12 (3) for ΔC-ASO, \*\*\*$P < 0.001$, ns, not significant, two-way ANOVA with Bonferroni's test). The error bars represent SEM.

G   *Pten^{ΔC/ΔC}* mice (2–4 months) treated with *Slc6a20a*-ASO display normal levels of climbing in the Laboras test. *n* = 6 mice for WT-saline, seven for WT-ASO, seven for ΔC-saline, and eight for ΔC-ASO, \*$P < 0.05$, \*\*$P < 0.01$, ns, not significant, two-way ANOVA with Bonferroni's test. The error bars represent SEM.

H   *Slc6a20a*-ASO treatment has no effect on the self-grooming of *Pten^{ΔC/ΔC}* mice (2–4 months) in the Laboras test. (*n* = 6 mice for WT-saline, seven for WT-ASO, seven for ΔC-saline, and eight for ΔC-ASO, ns, not significant, two-way ANOVA with Bonferroni's test). The error bars represent SEM.

Source data are available online for this figure.

(Appendix Fig S6A). We expressed these four constructs in heterologous cells (HEK293T) and measured concentration-dependent transport currents evoked by glycine and proline. For this experiment, we used the IonFlux system, an automated patch (auto-patch) clamp system that enables high-throughput and averaged measurements of ion fluxes from multiple (~ 20) cells.

All four SLC6A20 types (mouse/human, a/b) mediated largely similar levels of glycine and proline transports (Fig 7A; Appendix Fig S7A). Specifically, glycine and proline induced transport currents with comparable $K_{0.5}$ values (the concentrations of glycine and proline eliciting half-maximal transports; 19.7–63.16 μM and 19.64–156.8 μM for glycine and proline, respectively; $K_{0.5}$ for glycine, human V1 = 19.7 μM, human V2 = 30.98 μM, mouse A = 63.16 μM, mouse B = 35.8 μM; $K_{0.5}$ for proline, V1 = 156.8 μM, V2 = 49.03 μM, A = 19.64 μM, B = 44.83 μM).

In control experiments, the proline/glycine-induced currents for mouse SLC6A20A were confirmed to be sodium chloride (NaCl)-dependent, being markedly suppressed when sodium chloride was replaced with choline chloride or sodium gluconate (Fig 7B; Appendix Fig S7B). SLC6A20A could transport sarcosine (a positive control transport substrate) in a sodium chloride-dependent manner but not histidine or GABA (negative control substrates; Appendix Fig S8). In addition, proline/glycine-induced currents did not show current reversals at positive holding potentials, as shown by the experiments performed using single HEK293T cells expressing SLC6A20A (not multiple HEK293T cells) in a conventional patch-clamp (not auto-patch) setup (Fig 7C).

Our results suggest that SLC6A20 regulates extracellular glycine levels and NMDAR function in the mouse brain, implicating SLC6A20 in clinical applications. We thus tested whether SLC6A20 is expressed in the human brain. Immunoblot analysis of human neural progenitor cells (NPCs) and NPC-derived neurons (1–6 weeks) indicated that SLC6A20 proteins are clearly detected in both NPCs and NPC-derived neurons, with the expression levels slightly decreasing across the developmental stages (Fig 7D).

# Discussion

## PTEN C terminus and NMDAR function

Our data indicate that PTEN is important for the maintenance of normal excitatory synapse density and synaptic plasticity. The decreased number and function of excitatory synapses in *Pten^{ΔC/ΔC}* hippocampal DG and CA1 regions are supported EM and mEPSC results (Fig EV2). These results are in sharp contrast with the prevailing notion that PTEN negatively regulates excitatory synapse number, based on the previous studies on mice lacking *Pten* globally or in specific cell types. Specifically, mice carrying a homozygous deletion of *Pten* limited to subsets of differentiated neurons (exons 4–5; *Nse-Cre*) show dendritic hypertrophy and increased dendritic spine density in DG granule cells (Suzuki *et al*, 2001; Kwon *et al*, 2006). Another study using the same mouse line reports increased excitatory basal transmission in association with increased presynaptic function at DG, but not CA1, synapses at about 14–19 postnatal weeks, but not at 8–12 or 20–30 weeks (Takeuchi *et al*, 2013). Moreover, retroviral-mediated *Pten* deletion in newborn DG neurons (exon 5; Cre-expressing retrovirus) induces increases in dendritic spine and excitatory synapse density, as well as in excitatory synaptic transmission (Williams *et al*, 2015). Lastly, Purkinje cell-specific *Pten* deletion (exon 5; *L7-Cre*) induces an increase in parallel fiber EPSCs through postsynaptic changes (Cupolillo *et al*, 2016).

This difference between previous and our current results (increased vs. decreased excitatory synapse density and function) may be attributable to the difference between the complete loss of PTEN protein in the previous studies versus deletion of the PTEN C terminus in the current study. For example, targeting exons 4–5 of *Pten* in a previous study completely eliminates the PTEN protein, which induces the disinhibition of the PI3K-Akt-mTOR signaling pathway (Suzuki *et al*, 2001; Kwon *et al*, 2006). In contrast, our PTEN-PB deletion does not induce degradation of the truncated protein or

alter the phosphatase function of the protein, as supported by the normal amount of the PTEN protein and normal Akt and mTOR activity downstream of PTEN (Fig EV1). It is unclear how PTEN C-terminal deletion reduces excitatory synapse number in the $Pten^{ΔC/ΔC}$ hippocampus. A possible mechanism is the suppressed NMDAR function because NMDARs promote the formation and maturation of dendritic spines and excitatory synapses (Engert & Bonhoeffer, 1999; Maletic-Savatic et al, 1999; Hering & Sheng, 2001).

$Pten^{ΔC/ΔC}$ mice also show reduced LFS-LTD, consistent with previous results (Jurado et al, 2010). Surprisingly, our data further indicate that HFS-LTP is suppressed, whereas TBS-LTP is not (Fig 1). In addition, the NMDA/AMPA ratio is decreased in $Pten^{ΔC/ΔC}$ mice, whereas AMPAR-mediated basal transmission and paired-pulse facilitation are normal. Both direct activation of NMDARs by DCS (glycine-site co-agonist) and indirect activation of NMDARs by sarcosine (competitive GlyT1 antagonist) normalize HFS-LTP as well as LFS-LTD at $Pten^{ΔC/ΔC}$ SC-CA1 synapses (Fig 4).

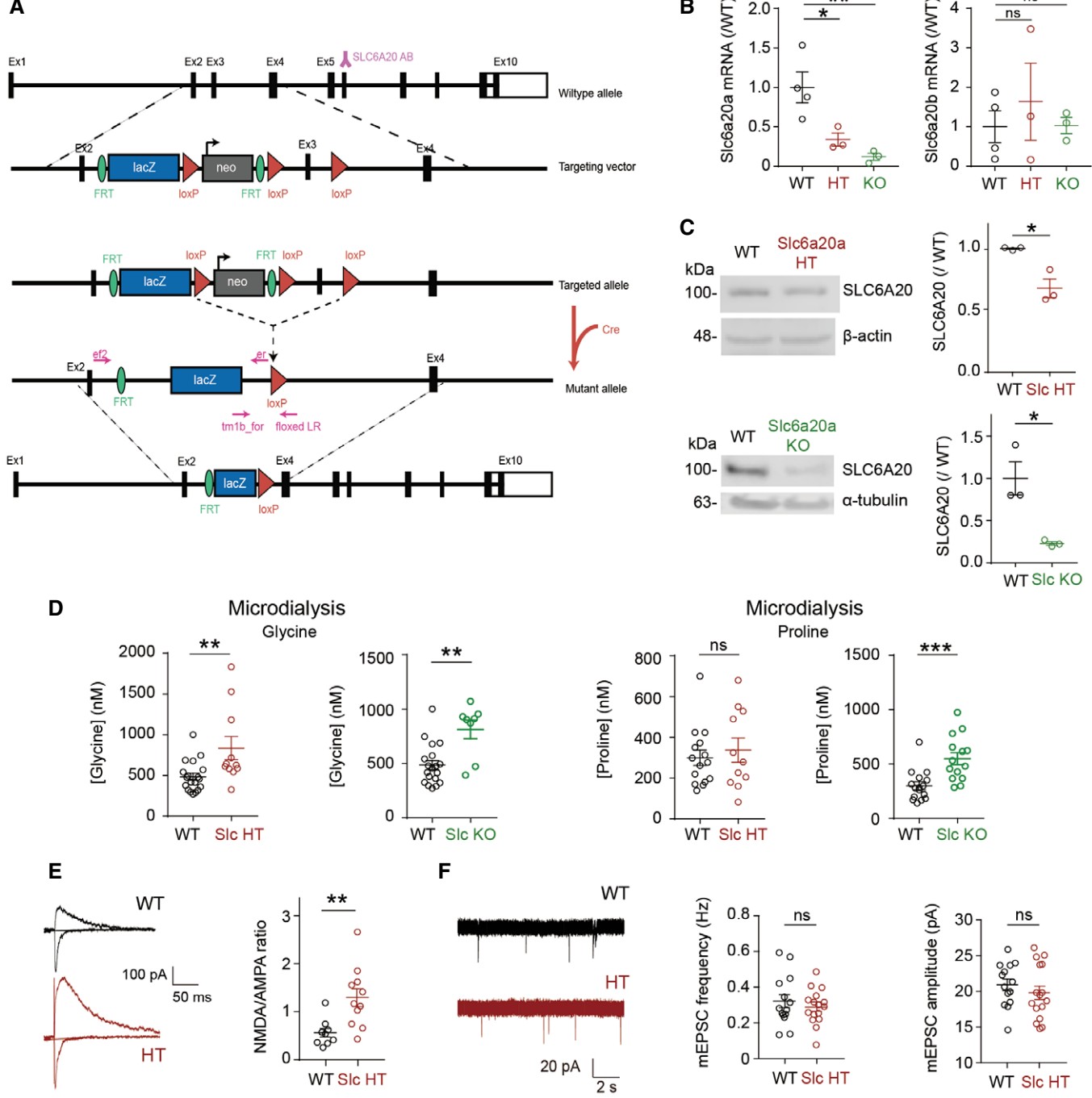

**Figure 6.**

Figure 6. *Slc6a20a*-mutant mice show abnormal increases in extracellular glycine levels and NMDAR function.

A   Schematic diagram showing the generation of *Slc6a20a*-mutant mice. The SLC6A20 antibody is denoted by "Y" above exon (Ex) 6.
B   Decreased levels of *Slc6a20a*, but not *Slc6a20b*, mRNAs in the brain of *Slc6a20a*$^{+/-}$ (HT) and *Slc6a20a*$^{-/-}$ (KO) mice (P21–28), as shown by qRT–PCR analysis. ($n$ = 4, 3, 3 mice for WT, HT, and KO; *$P$ < 0.05, **$P$ < 0.01, ns, not significant, one-way ANOVA with Bonferroni's test). The error bars represent SEM.
C   Reduced expression of SLC6A20 proteins in the brains of *Slc6a20a*$^{+/-}$ and *Slc6a20a*$^{-/-}$ mice (P21 for HT and P24–28 for WT and KO), as indicated by immunoblot analysis of total brain lysates. β-actin and α-tubulin were used as loading controls. ($n$ = 3 mice for WT, HT, and KO, *$P$ < 0.05, Student's *t*-test). The error bars represent SEM.
D   Increased extracellular levels of glycine and proline in the brain of *Slc6a20a*$^{+/-}$ and *Slc6a20a*$^{-/-}$ mice, as shown by microdialysis analyses. Note that glycine levels are increased in both *Slc6a20a*$^{+/-}$ and *Slc6a20a*$^{-/-}$ mice, whereas proline levels are increased only in *Slc6a20a*$^{-/-}$ mice. (glycine, $n$ = 19 mice for WT, 11 for HT, 8 for KO, **$P$ < 0.01, Mann–Whitney $U$-test; proline, $n$ = 15 mice for WT, 11 for HT, 14 for KO, ***$P$ < 0.001, ns, not significant, Mann–Whitney $U$-test). The error bars represent SEM.
E   Increased NMDA/AMPA ratio at hippocampal SC-CA1 synapses in *Slc6a20a*$^{+/-}$ mice (P17–20), as indicated by the ratio of NMDAR EPSCs to AMPAR EPSCs. ($n$ = 9 neurons from three mice for WT and 11 (4) for Slc6a20a HT mice, **$P$ < 0.01, Mann–Whitney $U$-test). The error bars represent SEM.
F   Normal frequency and amplitude of mEPSCs in hippocampal CA1 pyramidal cells in *Slc6a20a*$^{+/-}$ mice (P17–20). (mEPSC, $n$ = 14 neurons from three mice for WT and 16 (3) for HT; ns, not significant, Student's *t*-test). The error bars represent SEM.

Source data are available online for this figure.

These results collectively suggest that reduced NMDAR function may suppress LFS-LTD and HFS-LTP.

Glycine and D-serine act as co-agonists for extrasynaptic and synaptic NMDARs, respectively (Gray & Nicoll, 2012; Papouin *et al*, 2012). How does this fit with the findings of the present study? Our hypothesis is that the concentration of extracellular glycine is reduced by abnormally increased expression of *Slc6a20a*. If glycine is a specific co-agonist for extrasynaptic NMDARs, a reduction in ambient glycine concentration would selectively reduce extrasynaptic NMDAR-mediated currents. This is compatible with our finding because the evoked NMDAR currents measured in our NMDA/AMPA ratio experiments likely reflect both synaptic and extrasynaptic NMDAR activities, given that glutamate molecules spilled over from excitatory synapses would also activate extrasynaptic NMDARs by acting in concert with glycine (Papouin & Oliet, 2014). Our results are also similar to the increased evoked NMDAR currents observed in GlyT1-mutant mice (Tsai *et al*, 2004b; Gabernet *et al*, 2005).

The reduced NMDAR currents in *Pten*$^{ΔC/ΔC}$ SC-CA1 synapses might reflect a change in the ratio of synaptic and extrasynaptic NMDARs involving alterations in NMDAR-associated synaptic scaffolding proteins such as PSD-95, GKAP, and Shank. However, our data suggest that PTEN C-terminal deletion does not affect synaptic levels of NMDAR subunits (Fig EV1H). In addition, PTEN is not tightly associated with postsynaptic multi-protein complexes, being recruited to synapses in a regulated manner during synaptic plasticity or Aβ-induced synaptic weakening (Jurado *et al*, 2010; Knafo *et al*, 2016). Moreover, our results indicate that glycine concentrations are strongly correlated with NMDAR functions in multiple experimental conditions (untreated/ASO-treated *Pten*$^{ΔC/ΔC}$ mice and *Slc6a20a*$^{+/-}$ mice).

## SLC6A20A regulates brain glycine homeostasis and NMDAR function

Our data suggest that the reduced NMDAR function in *Pten*$^{ΔC/ΔC}$ mice involves increased expression of SLC6A20A (Fig 2). Specifically, our RNA-Seq transcriptomic analyses, performed in an unbiased manner using both DEG and GSEA analyses, indicate an abnormal increase in the expression of *Slc6a20a*, a change also validated by qRT–PCR and immunoblot analyses (Figs 2A and C, and EV1J).

With regard to the mechanisms underlying increased *Slc6a20a* expression in *Pten*$^{ΔC/ΔC}$ mice, our data suggest that the possibility that altered excitatory synaptic structure or function may liberate synaptic protein/actin-associated proteins such as β-catenin that can translocate to the nucleus and regulate gene expression. In line with this hypothesis, excitatory synapse number and synaptic plasticity are suppressed (Figs EV2 and 1), phosphorylation of β-catenin (Ser-675) is increased in the nucleus-enriched P1 fraction but not in whole-brain lysates of *Pten*$^{ΔC/ΔC}$ mice (Fig 2D), and β-catenin binding to the promoter region of the *Slc6a20a* gene is increased (Fig 2F). In addition, a previous study has shown that a mutant PTEN lacking the C-terminal PB (PDZ-binding) domain destabilizes adherens junctions in retinal epithelial cells and promotes nuclear localization of β-catenin to regulate TCF/LEF-dependent gene expression (Kim *et al*, 2008). In neurons, β-catenin associates with synaptic N-cadherin and regulates excitatory synaptic structure and function in an activity-dependent manner (Yu & Malenka, 2003; Tai *et al*, 2007), In addition, the deletion of Shank3, an abundant excitatory postsynaptic scaffolding protein (Sheng & Kim, 2011), can induce synapse-to-nuclear translocation of β-catenin and alter transcriptional activity through HDAC2 expression (Qin *et al*, 2018).

Previous studies have shown that full-length PTEN can localize to the nucleus to regulate diverse cellular processes in neuronal and non-neuronal cells (Gil *et al*, 2007; Planchon *et al*, 2008; Howitt *et al*, 2012; Bassi *et al*, 2013; Zhang *et al*, 2013; Goh *et al*, 2014; Fricano-Kugler *et al*, 2018; Igarashi *et al*, 2018). For instance, ischemic or excitotoxic neuronal injuries can induce the nuclear localization of PTEN in a ubiquitin-dependent manner to regulate nuclear signaling pathways and neuronal survival or death (Howitt *et al*, 2012; Zhang *et al*, 2013; Goh *et al*, 2014). Therefore, the PB-lacking mutant PTEN protein that can no longer be stabilized at synaptic sites, or at subcellular sites of non-neuronal cells such as those in the meninges, choroid plexus, and capillaries, might abnormally translocate into the nucleus, altering the expression of certain genes in the nucleus, such as *Slc6a20a*. However, a previous study has shown that a mutant PTEN lacking the last three aa residues show unaltered nuclear localization (Gil *et al*, 2006), suggesting that the enhanced expression of *Slc6a20a* is an unlikely consequence of PTEN-PB deletion.

*Slc6a20a* encodes a proline transporter. Previous *in vitro* studies have reported that SLC6A20A/B preferentially transports imino acids, including proline, hydroxyproline, and betaine, although it

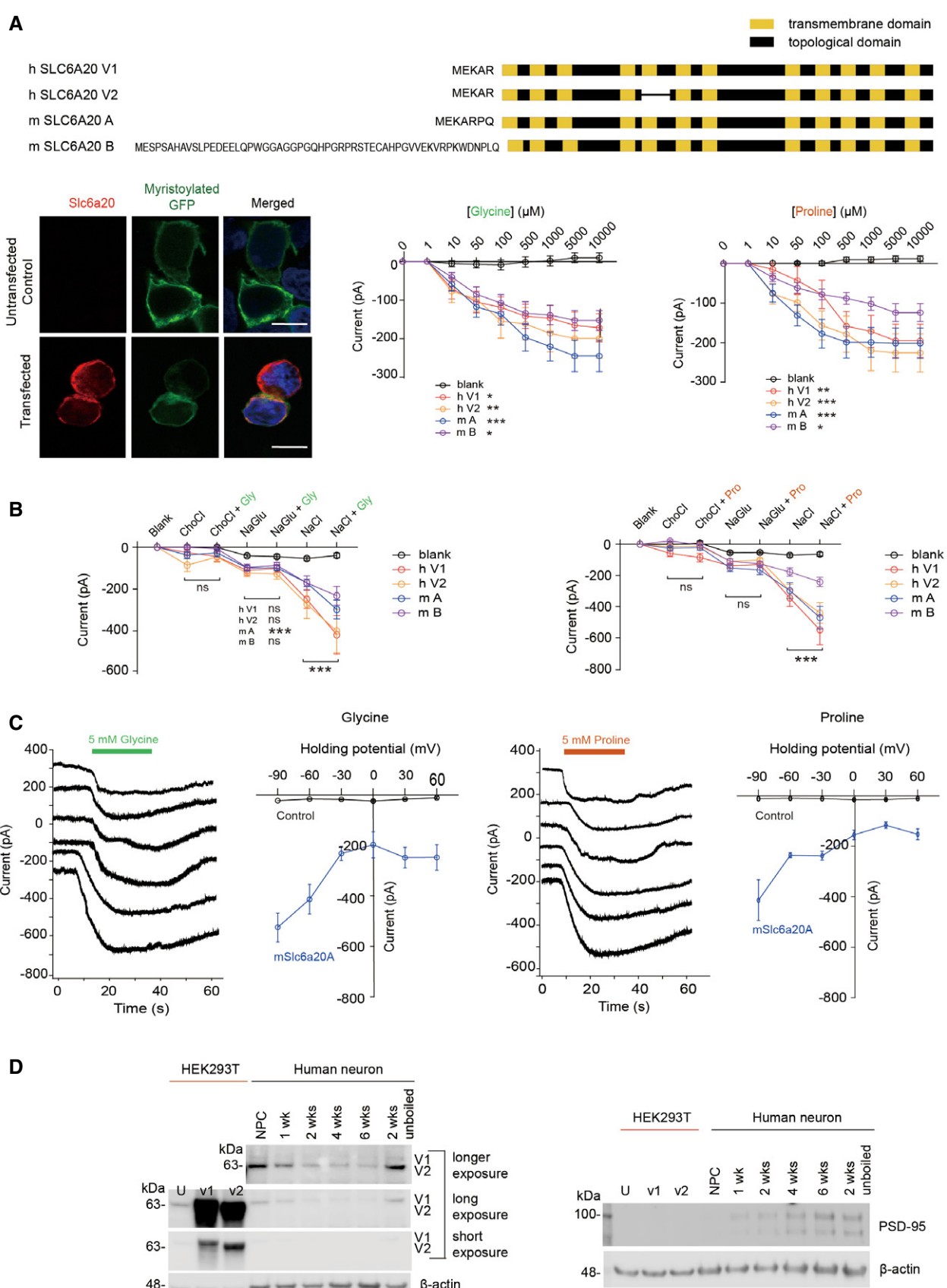

**Figure 7.**

**Figure 7.  Mouse and human SLC6A20 proteins mediate proline and glycine transports at similar levels, and human neurons express SLC6A20 proteins.**

A  All four SLC6A20 protein variants (human SLC6A20 v1 and v2; mouse SLC6A20A and SLC6A20B; shown in schematic diagrams) mediate similar levels of glycine and proline transports. HEK293T cells expressing SLC6A20 protein variants were used to measure ensemble currents induced by increasing concentrations of glycine/proline in auto-patch experiments. Sample images indicate HEK293T cells expressing human SLC6A20 v1; myristoylated GFP expression, used only for this imaging but not for transport measurements, show detectable surface expression of expressed SLC6A20 proteins. Scale bar, 10 μm. Increasing concentrations of glycine/proline were sequentially applied to HEK293T cells, as shown in Appendix Fig S7A. The indicated currents are average values from an ensemble of multiple (~ 20) cells in a single well. (glycine, $n = 34$ cells for untransfected/blank HEK293T cells, 36 for human SLC6A20-V1, 38 for human SLC6A20-V2, 48 for mouse SLC6A20A, 51 for mouse SLC6A20B, *$P < 0.05$, **$P < 0.01$, ***$P < 0.001$ (relative to untransfected), two-way ANOVA with Tukey's test; proline, $n = 46$ cells for non-transfected, 43 for hSLC6A20-V1, 31 for hSLC6A20-V2, 54 for mSLC6A20A, 54 for mSLC6A20B, *$P < 0.05$, **$P < 0.01$, ***$P < 0.001$ relative to untransfected, two-way ANOVA with Tukey's test). The error bars represent SEM.

B  Glycine/proline-induced currents for mouse SLC6A20A are dependent on sodium chloride, as shown by the suppression of the currents when sodium chloride is replaced with choline chloride or sodium gluconate. Each experimental condition was sequentially applied to HEK293T cells, as shown in Appendix Fig S7B. The indicated currents are average values from an ensemble of multiple (~ 20) cells in a single well. (glycine, $n = 21$ cells for untransfected/blank, 17 for human SLC6A20-V1, 25 for human SLC6A20-V2, 33 for mSLC6A20A, 28 for mSLC6A20B ***$P < 0.001$ (relative to buffer not containing glycine), two-way ANOVA with Bonferroni's test; proline, $n = 17$ cells for untransfected, 23 for human SLC6A20-V1, 47 for SLC6A20-V2, 31 for mSLC6A20A, and 31 for mSLC6A20B, ***$P < 0.001$, two-way ANOVA with Bonferroni's test). The error bars represent SEM.

C  Glycine/proline-induced currents for mouse SLC6A20A do not show reversal potentials, as shown by the lack of reverse currents induced by increasing holding potentials in HEK293T cells. These experiments were performed using single HEK293T cells in a conventional patch-clamp (not auto-patch) setup. (glycine, $n = 3, 7, 3, 3, 3,$ and 3 cells for untransfected HEK293T cells, 10, 11, 5, 6, 6, and 5 cells for mouse SLC6A20A, at $-90, -60, -30, 0, 30,$ and 60 mV of holding potentials, respectively; proline, $n = 4, 7, 4, 5, 3,$ and 4 cells for untransfected HEK293T cells, 3, 6, 3, 3, 3, and 3 cells for mouse SLC6A20A, at $-90, -60, -30, 0, 30,$ and 60 mV of holding potentials, respectively). The error bars represent SEM.

D  Human neural progenitor cells (NPCs) and neurons differentiated from NPCs for 1–6 weeks express SLC6A20 proteins, as revealed by immunoblot analysis using the SLC6A20 antibody directed against a region of human SLC6A20 (aa 301–369; KATFNYENCLKKVSLLLTNTFDLEDGFLTASNLEQVKGYLASAYPSKYSEMFPQIKNCSLESELDTAVQ; Sigma) that is 83% identical to the mouse SLC6A20 sequence. Two different splice variants of human SLC6A20 (v1 and v2) expressed in HEK293T cells were used as positive controls. PSD-95, an abundant excitatory postsynaptic scaffold protein, was used as a maker for neuronal maturation, and β-actin was used as a loading control. Note that the band intensity of "unboiled" SLC6A20 at 2-week time point is stronger relative to the boiled SLC6A20 band at 2 weeks, in line with that SLC6A20 is a transmembrane protein with twelve transmembrane domains that easily forms larger protein aggregates upon boiling and get stuck at the start location of western gels, decreasing the protein band intensity of SLC6A20. We did not try the unboiling of other samples at different time points because one sample was enough for proof of concept. U, untransfected HEK293T cells; V1/V2, HEK293T cells expressing human SLC6A20-v1/v2.

Source data are available online for this figure.

also transports amino acids, including phenylalanine, valine, cysteine, glycine, and glutamate, at lower efficiencies (Kowalczuk *et al*, 2005; Takanaga *et al*, 2005). Intriguingly, our data associate increased *Slc6a20a* expression in *Pten$^{\Delta C/\Delta C}$* mice with decreased extracellular proline/glycine levels (Fig 3A–C). In addition, *Slc6a20a* deletion in mice leads to stronger increases in extracellular levels of glycine relative to proline (Fig 6D). More directly, both human and mouse SLC6A20 proteins mediate proline and glycine transports at comparable levels (Fig 7A–C; Appendix Fig S7).

A previous study by Kowalczuk *et al* reported that mouse SLC6A20A, widely expressed in multiple tissues including the brain, mediates strong proline transport but weak glycine transport. This difference may be attributable to that they used Xenopus oocytes, whereas we used HEK293T cells (human embryonic kidney cells), which may provide different cellular environments for protein expression and post-translational modifications. In support of this possibility, the GAT-3 GABA transporter has been shown to transport β-alanine differentially in Xenopus oocytes and mammalian cells; specifically, GAT-3 can transport both GABA and β-alanine efficiently (~Km of 18 and 28 μM) in Xenopus oocytes (Liu *et al*, 1993b), whereas the same GAT-3 can transport GABA but not β-alanine in HeLa cells (Clark *et al*, 1992). Yet, GAT-3 can transport β-alanine and be inhibited of GABA transport by β-alanine when stably expressed in LLC-PK$_1$ cells, a mammalian cell type (Clark & Amara, 1994). In addition, the concentration of glycine in the brain is much greater than that of proline (i.e., > 10-folds in humans (Jones *et al*, 2006)). It should be pointed out, however, that extracellular levels of proline are also decreased in *Pten$^{\Delta C/\Delta C}$* mice, although whole-brain proline levels were not changed, which could be attributable to that SLC6A20A proteins may transport

glycine out of the brain in addition to brain cells, whereas they transport proline more strongly into brain cells. Although further details remain to be determined, our results suggest that SLC6A20 proteins act as the regulator of both proline and glycine homeostasis in the brain.

Importantly, *Slc6a20a* is highly expressed in various brain regions, including the choroid plexus and meninges, as supported by our FISH and X-gal staining results (Fig EV3–5) and previous *in situ* hybridization results (Kowalczuk *et al*, 2005; Takanaga *et al*, 2005; Dahlin *et al*, 2009). In addition, *Slc6a20a* is highly expressed in specific cell types, including astrocytes and microglia, but only modestly (glutamate neurons) or minimally (GABAergic neurons) expressed in others (Fig EV4 and EV5). These results are in line with previous microarray results showing that *Slc6a20a/b* expression is detected in choroid plexus epithelial cells, but not in the blood–brain barrier (BBB) endothelial cells (Saunders *et al*, 2013).

Epithelial cells in the choroid plexus serve as a barrier between the blood and cerebrospinal fluid (CSF), a site of CSF production and secretion, and a gatekeeper that regulates the blood–CSF transport of small molecules through specific transporters (Saunders *et al*, 2008; Engelhardt & Sorokin, 2009; Saunders *et al*, 2013). The outer cells in the meninges also function as a barrier between blood and CSF, and regulate various transports (Saunders *et al*, 2008; Engelhardt & Sorokin, 2009). Astrocytes also rapidly take up and degrade glycine, producing and releasing serine and lactate (Verleysdonk *et al*, 1999). In addition, it has been shown that brain-to-blood efflux exists for various neurotransmitters and small molecules (Takanaga *et al*, 2002; Ohtsuki, 2004).

These observations collectively suggest the possibility that an increase in *Slc6a20a* expression at sites of blood–CSF barriers, such

as choroid plexus and meninges, or in metabolically active astrocytes and microglia might alter the transport or clearing of imino and amino acids, including proline and glycine, and alter the concentration of these molecules in the whole brain or in local environments around synapses. Our data seem to support both hypotheses (glycine regulation around the synapse) because (i) ASO-mediated suppression of *Slc6a20a* expression normalizes whole-brain glycine levels, NMDAR function, and repetitive climbing in $Pten^{\Delta C/\Delta C}$ mice (Fig 5) and (ii) $Slc6a20a^{+/-}$ and $Slc6a20a^{-/-}$ mice display increased levels of extracellular glycine/proline and NMDAR function in the brain (Fig 6).

Although further details remain to be elucidated, our results at minimum suggest that (i) SLC6A20A is a novel regulator of brain glycine and proline homeostasis; (ii) SLC6A20A could underlie abnormal synaptic functions and behaviors; (iii) *Slc6a20a* expression can be altered by defects in brain disease-related proteins, such as PTEN; and (iv) SLC6A20A could be a potential target for intervention in brain disorders associated with lowered NMDAR function, including schizophrenia and ASD (Olney *et al*, 1999; Lee *et al*, 2015). Importantly, *SLC6A20*, in addition to *SLC6A5* (GlyT2), has been implicated in schizophrenia in recent exome-sequencing studies (Purcell *et al*, 2014) (see also http://atgu.mgh.harvard.edu/~spurcell/genebook/genebook.cgi?user=guest&cmd=overview). In addition, we normalized the phenotypes of $Pten^{\Delta C/\Delta C}$ mice using antisense oligonucleotides (Fig 5), a clinically more relevant preparation (Rinaldi & Wood, 2018). Moreover, SLC6A20 proteins could be detected in human NPCs and NPC-derived neurons (Fig 7D).

Lastly, human mutations in *SLC6A20*, encoding the SLC6A20 protein that can mediate proline transport, have been linked to complex digenic iminoglycinuria (a renal disorder with impaired reabsorption of glycine and imino acids [proline and hydroxyproline]) and hyperglycinuria (Broer *et al*, 2008). In addition, known glycine transporters such as GlyT1 and GlyT2 encoded by SLC6A9 and SLC6A5, respectively, are associated with glycine encephalopathy (Alfadhel *et al*, 2016; Kurolap *et al*, 2016). GlyT1- and GlyT2-mutant mice also show early postnatal lethality associated with impaired glycinergic transmission and phenotypes mimicking human glycine encephalopathy and hyperekplexia (Gomeza *et al*, 2003a; Gomeza *et al*, 2003c; Eulenburg *et al*, 2010). It is thus possible that $Slc6a20a^{-/-}$ mice may also display impaired renal glycine/proline transport or glycinergic transmission. However, WT, $Slc6a20a^{+/-}$, and $Slc6a20a^{-/-}$ mice show a Mendelian ratio of 1.00:1.93:0.55, unlike the early postnatal lethality of GyT1- and GlyT2-null mice (Gomeza *et al*, 2003a; Gomeza *et al*, 2003c). In addition, SLC6A20A proteins are detected in various brain areas, including cortical and hippocampal regions (Fig EV3B), contrary to GlyT1 and GlyT2 that are more strongly expressed in brain stem areas (Gomeza *et al*, 2003b; Betz *et al*, 2006; Harvey & Yee, 2013), where glycinergic transmission critically regulates respiratory and motor function. These aspects led us not to test glycinergic transmission in the brain stem areas of $Slc6a20a^{-/-}$ mice in the present study, although it should be directly tested in future studies.

In conclusion, our results indicate that PTEN-PB deletion leads to marked decreases in excitatory synapse number and NMDAR-dependent synaptic transmission and plasticity that are associated with repetitive climbing behavior in mice, and reveals a novel role

of SLC6A20A in the regulation of brain glycine homeostasis and NMDAR function.

# Materials and Methods

### Animals

$Pten^{\Delta C/\Delta C}$ mice ($Pten^{tm(Q399stop)amc}$ mice) have been previously reported (Knafo *et al*, 2016). Mouse genotyping was performed by polymerase chain reaction (PCR) using the following primers: 5′-AAG TGG CTG AAG AGC TCT GA - 3′ and 5′- CAG CCA ATC TCT CGG ATG TC-3′. The expected size of mutant PCR band was 460 bp, while WT band was 160 bp. *Slc6a20a*-mutant mice were purchased (EMMA ID: 09248, *Slc6a20a tm1b(KOMP)Wtsi*) as sperms and generated through *in vitro* fertilization. Tm1b allele is reporter-tagged deletion allele (post-Cre). Exon3 was deleted by creating a frame-shift using Cre recombination. The neo selection cassette was removed, while LacZ reporter cassette was kept to visualize the gene expression at the protein level. The genotyping primers used are as follows: for WT alleles: Slc6a20a_Ef2 5′CCAAGGTCAACAGCTCCGTCCA3′, Slc6a20a_Er 5′GAGAAGTGCC AACCATATGCCTAATGC3′ and for mutant tm1b_forw 5′CGGTCGC TACCATTACCAGT3′ and Floxed LR 5′ACTGATGGCGAGCTCAG ACC3′. Both of these mice strains were maintained according to the Requirements of Animal Research at KAIST, and experimental procedures were approved by the Committee on Animal Research at KAIST (KA2012-19 and KA2020-89).

### RNA-Seq library preparation and sequencing

P21 mouse brains were dissected carefully and soaked in RNAlater solution (Ambion, AM7020) to stabilize RNA. RNA extraction, library preparation, cluster generation, and sequencing were conducted by Macrogen Inc. (Seoul, Korea). RNA samples were prepared for sequencing using a TruSeq RNA Sample Prep Kit v2 (Illumina) according to the manufacturer's instructions. Sequencing was carried out using an Illumina's HiSeq 4000 to generate 101-bp paired-end reads. Image analysis and base calling were performed with the standard Illumina software RTA (Real-Time Analysis v2.7.3). The BCL (base calls) binary was converted into FASTQ utilizing Illumina package bcl2fastq (v2.17.1.14). An approximate average of 69 million reads per replicate was obtained. Raw data were submitted to the GEO (Gene Expression Omnibus) repository with accession GSE119236.

### RNA-Seq analysis

The raw reads were pre-processed using Trimmomatic (Bolger *et al*, 2014) (version 0.35, options: LEADING:3 TRAILING:3 MAXINFO:80:0.4 MINLEN:36) and the trimmed reads were aligned to the *Mus musculus* genome (GRCm38) using TopHat2 (version 2.1.0) with default parameters (Kim *et al*, 2013). The gene-level read counts were calculated from the aligned reads by using HTSeq (Anders *et al*, 2015). Further normalization and differential gene expression analysis were carried out using R/Bioconductor DESeq2 package (Love *et al*, 2014). The *P*-values obtained by the Wald test were first corrected by applying empirical estimation of the null

distribution using the R fdrtool (v.1.2.15) package and then were adjusted for multiple testing with the Benjamini–Hochberg correction. Genes with an adjusted FDR value < 0.1 were considered to be differentially expressed.

## Gene set enrichment analysis

The GSEA tool (gsea2-2.2.4.jar) (Subramanian *et al*, 2005) was used to determine whether *a priori*-defined gene sets showed statistically significant differences between $Pten^{\Delta C/\Delta C}$ and wild-type mice. Specifically, GSEA analysis was performed using GSEA Pre-ranked module on gene set C5 category downloaded from Molecular Signature Database (MSigDB) v6.0 (http://software.broadinstitute.org/gsea/msigdb) and the input rankings of all expressed genes were based on the sign of the fold change multiplied by the inverse of the *P* value. GSEA calculates an enrichment score (ES), which reflects the degree to which a gene set is overrepresented at the highly up- (positive ES) or downregulated (negative ES) gene lists. NES, normalized enrichment score (ES), is the primary statistics for gene set enrichment results and calculated for each gene set based on permutation. All the recommended default settings with 1,000 permutations and a classic scoring scheme were used, and the gene sets with FDR (*q*-value) < 0.05 were considered significantly enriched.

## qRT–PCR

To quantify mRNA levels, total brain RNAs were extracted, and cDNAs were synthesized using TOPscript™ cDNA synthesis kit (Enzynomics, EZ005) according to the manufacturer's instructions. SsoAdvanced™ SYBR® Green Supremix (BIORAD, 1725260) and CFX96™ Real-Time System were used for real-time PCR. The following primers were used:

Omd For: 5′cacactgagccatctcaccagc3′, Rev: 5′ggctcttggtcataatcgt-catcccca3;

Bst2 For: 5′gaacaccacgcacctgttgca3′ , Rev: 5′ctccagggcttgagacac-cttct3′;

Nkx2-1 For: 5′ ggggacgtgagcaagaacat3′, Rev: 5′ tagcctg-gcgcttcatcttg3′;

Cep295nl For: 5′ggctcagtcatctggagacgca3′, Rev: 5′ggctggcatctc-taacggcttct3′;

Bche For: 5′gcagactcagcataccaaggt3′, Rev: 5′ ggataccgagaaag-gcagtca 3′;

Fezf1 For: 5′ ccaaatgccaagcccaaagtt 3; Rev: 5′ tcccacacactttaca-cacga 3′;

Hmcn2 For: 5′cctctattcctgccaggctgaga3′, Rev: 5′caatgtca-catcagcaccagcca3′;

Slc6a20a (for aso knockdown validation, Fig 5B):

For: 5′ ctcacgggaacactgcagtacca3′, Rev: 5′ gatgaagccaccagcaaacc-gat3′;

Slc6a20a (for slc6a20a mRNA expression, Fig 6B):

For: 5′ ttctcactgggcttgggtttt 3′, Rev: 5′ ggcaaatatggaggtggagct3′;

Slc6a20a (for PTEN399 RNA-seq validation, Fig EV1):

For: 5′cagcaaggtcatctccagctacct3′, Rev: 5′ccatggtgaacaccatgccaa-ca3′;

Slc6a20b For: 5′ gtaccaagcctgggatgctactca 3′, Rev: 5′ ctgatcttga agtggcgtgtgaca 3′;

Fam167 For: 5′ ggagagaagggtgaggaggat 3′, Rev: 5′cactcaagcgca-gaatccatg3′;

Gapdh For: 5′tcagcaatgcatcctgcaccacc3′, Rev: 5′tggcagtgatggcatg-gactgtg3′.

## Chromatin immunoprecipitation analysis

Brain tissues (50 mg) was collected, cross-linked with 1% formalde-hyde for 10 min, and quenched by the addition of glycine (at the final concentration of 0.125 M) for 5 min before freezing at −80°C. The chromatin was extracted by the 1% SDS lysis buffer and was subsequently sheared using an S220 Focused-ultrasonicator (Covaris) while still in the 1% SDS lysis buffer. After centrifugation, ∼1% of the supernatant was saved to use as input controls. The supernatant was incubated with (i) antibodies against β-catenin (BD bioscience, 5 μl per reaction) and (ii) sepharose A + G agarose 50% slurry overnight at 4°C under constant rotation. After washing five times, the bound complex was eluted from the beads by incubating with 250 μl elution buffer twice at room temperature. After reversing cross-links in 65°C overnight, proteins and RNA were removed using proteinase K (Invitrogen) and RNase (Roche), respectively. Then, the immunoprecipitated DNA and input DNA were purified by phenol/chloroform extraction. The quantification of ChIP signals was calculated as percent input. Primers targeting multiple sites in the region upstream of TSS (transcriptional start site) were designed and pretested in both input and ChIP samples. Only those with solid signals in both input and ChIP samples were chosen for the experiment. Purified DNA was subjected to qPCRs with primers against mouse *slc6a20* promoter. P1 (forward, −454 to −474 bp relative to TSS, 5′-TCCCAGAGGAGAGATGACTTT-3′; reverse, −543 to −562 bp relative to TSS, 5′-CCCATGTGTGCTAGGATCAA-3′); P2 (forward, −1,469 to −1,491 bp relative to TSS, 5′-GATTTGCATTTG-TATCCCTGCTC-3′; reverse, −1,582 to −1,602 bp relative to TSS, 5′-GAATACCTCTCCTGCCCAAAC-3′); P3 (forward, −1,597 to −1,618 bp relative to TSS, 5′- GTATTCAGCTCAGTGGGTAGAG-3′; reverse, −1,667 to −1,687 bp relative to TSS, 5′- CCACCATACA-CACCCAGTTAC -3′); P4 (forward, −3,051 to −3,071 bp relative to TSS, 5′-CTCTAACCTGACAAGCTCCAC-3′; reverse, −3,133 to −3,155 bp relative to TSS, 5′-TCAGTGTCCACACACTACATTAC-3′).

## Microdialysis

Mice were anesthetized with ketamine and xylazine (0.0275 μg/g or 27.5 μg/kg) and mounted in a stereotaxic frame (Kopf). After exposing the skull and drilling a hole, a CMA 8 Guide Cannula (CMA Microdialysis) was inserted in the hippocampus (AP: −1.6 mm; ML: −1.6 mm from bregma; DV, −1.2 for cannula and −1.7 for probe). In addition, additional screws were implanted in the skull to serve as an anchor. The protruding part of the CMA 8 Guide Canula, along with the anchor screws, were fixed with Zinc Polycarboxylate dental cement. One week after the guide cannula insertion, mice were anesthetized again, and CMA 8 Elite Microdialysis Probes (membrane diameter 0.5 mm, length 1 mm; stainless-steel shaft diameter 0.4 mm) was inserted through the guide cannula. The probe was connected to a microinjection pump (CMA Microdialysis) with polyethylene tubing (PE 50) and FEP tubing (INSTECH). Then, the probe was perfused with Perfusion Fluid CNS (CMA Microdialysis) into the inlet of the probe at a flow rate of 2 μl/min. Perfusates from the outlet of the tubing were automatically collected in plastic vials at 6°C using CMA 470 refrigerated fraction collector. Appropriate

volumes of dialysates were collected (100 and 50 µl, respectively) for the analysis of glycine and proline concentrations. Dialysates were stored at −20°C and then analyzed using Mass spectrometry and HPLC and Liquid chromatography and mass spectrometry (LC-MS/MS).

**Glycine measurement using HPLC**

Briefly, the derivatization was performed by mixing 100 µl *in vivo* microdialysate or standard solutions in 5% trichloroacetic acid (TCA) in 1:1 ratio. The mixture was centrifuged at 15,520 *g* for 15 min, after which only the supernatant was taken and syringe filtered (0.2 µm) before being subjected to analysis. The analysis used a Hitachi L-8900 Automatic Amino Acid Analyzer (Hitachi High-Technologies Corporation, Tokyo, Japan) with a 4.6 mm (ID) × 60 mm ion exchange column (Hitachi High-Technologies Corporation) in the Center for Research Facilities at Chungnam National University. Detection was by ninhydrin postcolumn derivatization (Wako Ninhydrin Coloring Solution Kit for HITACHI). An external standard was used to calculate the concentration of each amino acid.

**Proline measurement using LC-MS/MS**

The quantitative analysis of proline in CSF samples was performed by high performance liquid chromatography combined with hybrid triple quadrupole mass spectrometry (QTRAP 4000 QTRAP AB SCIEX, Foster City, CA, USA). Proline was separated by linear gradient elution at 300 µl/min using an amide column (50 mm × 2.1 mm i.d., 3.5 µm; Waters, USA). The mobile phase was composed of 0.1% formic acid in water (eluent A v/v) and 0.1% formic acid in acetonitrile (eluent B v/v) according to the following protocol: starting at 5% A from 0 to 0.01 min, increasing to 60% A at 4.0 min, remaining at 60% A to 4.1 min, and then re-equilibrating to the initial condition from 4.2 to 7 min. Multiple reaction monitoring (MRM) mode by electrospray positive ionization (ESI+) was used for quantification of proline according to the ionic transition $m/z$ 116.0 → 70.1 and 98.3.

**Slice electrophysiology**

Mice (P17–23, P19–37 for ASO) were anesthetized using isofluorane (Terrell), and brains were surgically prepared after carefully removing the skull. Sagittal hippocampal slices (300 µm) were prepared using a vibratome (Leica VT1200) and ice-cold sucrose-based artificial cerebrospinal fluid (sCSF) buffer containing (in mM): 212 sucrose, 10 D-glucose, 25 $NaHCO_3$, 5 KCl, 1.25 $NaH_2PO_4$, 1.25 L-ascorbic acid, 2 Na-pyruvate, 3.5 $MgSO_4$, and 0.5 $CaCl_2$ bubbled with 95% $O_2$ and 5% $CO_2$. The slices were recovered in a chamber while submerged in artificial cerebrospinal fluid buffer (aCSF) held at 32°C, containing (in mM): 125 NaCl, 10 D-glucose, 25 $NaHCO_3$, 2.5 KCl, 1.25 $NaH_2PO_4$, 1.3 $MgCl_2$, and 2.5 $CaCl_2$ for 30 min and subsequently recovered at room temperature for 30 min while being bubbled with 95% $O_2$ and 5% $CO_2$ through the entirety of the process.

For whole-cell patch recording, borosilicate glass pipettes (Harvard Apparatus) were pulled with a micropipette puller (Narshige). To record CA1 pyramidal cells, recording pipettes (3–4 MΩ) were filled with the following internal solutions: (i) for EPSC experiments (in mM): 117 $CsMeSO_4$, 10 TEA-Cl, 8 NaCl, 10 HEPES, 5 QX-314-Cl, 4 Mg-ATP, 0.3 Na-GTP, 10 EGTA with pH 7.3, and 285–300 mOsm, and (ii) for IPSC experiments (in mM): 115 CsCl2, 10 TEA-Cl, 8 NaCl, 10 HEPES, 5 QX-314-Cl, 4 Mg-ATP, 0.3 Na-GTP, 10 EGTA with pH 7.3, and 285–300 mOsm.

Data were filtered at 2 kHz and digitized at 10 kHZ using Multiclamp 700B and 1440 Digitizer (Molecular Devices). Series resistance was monitored in each sweep by measuring the peak amplitude of capacitance currents in response to short hyperpolarizing step pulse (5 mV, 40 ms). The acquired data were analyzed using Clampfit 10 (Molecular Devices).

Brain slices at P17–23 were used to measure miniature currents while holding at −70 mV. For mEPSC measurements, picrotoxin (100 µM) and tetrodotoxin (10 µM) were added to aCSF to block action potentials and inhibitory currents, respectively. For mIPSC measurements, NBQX (100 µM), AP5 (100 µM), and tetrodotoxin (TTX; 10 µM) were added to block AMPAR-mediated currents, NMDA-mediated currents, and action potentials, respectively. For spontaneous miniature recordings—sEPSC and sIPSC—similar process as the miniature counterparts mEPSC and mIPSC were followed, except for TTX, which was not added to allow for network activity and modulation of synaptic transmission.

For NMDA/AMPA ratio, picrotoxin (100 µM) was added to block $GABA_A$ receptor-mediated currents from slices at P17–21 (P20–37 for ASO). CA1 somatic cells were voltage clamped at −70 mV to measure AMPAR-mediated EPSCs, which were evoked every 15 s by the stimulation of stratum radiatum (SR) dendritic field. After obtaining a stable baseline, 30 consecutive responses were recorded as AMPAR components. The holding potential was then changed to +40 mV on the same neuron to measure NMDAR-mediated EPSCs. The NMDA component was determined by measuring the amplitude 60 ms after the stimulation. The ratio was calculated by dividing the average of 30 NMDAR EPSCs (peak amplitudes) by the average of 30 AMPAR EPSCs.

For extracellular field recordings, both stimulating and recording pipettes were filled with the aCSF solution and recorded at the stratum radiatum (SR) of the hippocampal CA1 region by stimulating the axon fibers of Schaeffer collateral from CA3. To induce HFS-LTP, high-frequency stimulation (100 Hz, 1 s) was applied after a stable baseline was acquired. For TBS-LTP, after acquisition of a stable baseline for 20 min, slices (4–6 weeks) were stimulated with 10 trains of four pulses at 100 Hz and responses were recorded for 1 h after the stimulation. For NMDA-dependent LTD, we added picrotoxin (100 µM) to aCSF and used slices at P16–22. After a stable baseline for 20 min, we stimulated the slices with low-frequency stimulation (1 Hz, 900 pulses for 15 min) followed by 1-h measurements of responses. For mGluR-dependent LTD, after 20-min stable baseline, we bath applied DHPG (50 µM) in aCSF to induce LTD for 10 min and recorded the responses for 1 h in the presence of picrotoxin (100 µM). The average rise slopes of fEPSPs during the last 10 min were compared. For input–output experiments, input was defined as the peak amplitude of the fiber volley, and the output was defined as the initial slope of fEPSP. The stimulation intensity ranged from 5 to 35 µA with 2.5-µA increments per minute. Paired-pulse facilitation was measured as by evoking two fEPSPs with inter-stimulus intervals ranging from 25 to 300 ms, and the ratios were calculated by dividing the initial slope of the second

fEPSP by that of the first fEPSP. For LTP rescue experiments, 20 μM DCS and 750 μM sarcosine were used. For LTD rescue experiments, 10 μM DCS or 750 μM sarcosine were used.

## Immunohistochemistry

Mice were transcardially perfused using cold 1% heparin followed by 4% paraformaldehyde-based fixation. Coronal slices (40 μm) were prepared using vibratome (Leica) and washed with 1× PBS for three times. After the 30 min of permeabilization (0.1% Triton-X) and 1 h of blocking (0.1% Triton-X/10% appropriate serum, in which the 2nd antibodies were made), primary antibody incubation was performed using NeuN (Millipore, MAB377) in 0.1% Triton-X PBS solution at 4°C for overnight. Next day, after washing three times with 1× PBS, secondary antibody incubation was performed using antibodies coupled to FITC (1:500), in 0.1% Triton-X in PBS at room temperature for 1 h. After washing three times with 1× PBS, the slices were incubated on cover slides with fluoroshield (Sigma) for DAPI staining for cell body staining. Z-stacked images were acquired using a confocal microscope (LSM 780, Zeiss). Slices were kept under the condition of cryo-protectant solution which contained 30% sucrose and 30% ethylene glycol in PBS for future use.

## Fluorescent *in situ* hybridization

Frozen sections (14 μm thick) were cut coronally through the hippocampal formation. Sections were thaw-mounted onto Superfrost Plus Microscope Slides (Fisher Scientific 12-550-15). The sections were fixed in 4% paraformaldehyde for 10 min, dehydrated in increasing concentrations of ethanol for 5 min, and finally air-dried. Tissues were then pre-treated for protease digestion for 10 min at room temperature. For RNA detection, incubations with different amplifier solutions were performed in a HybEZ hybridization oven (Advanced Cell Diagnostics) at 40°C. In this study, we used fluorescent probes to label Slc6a20, PTEN, Slc6a9 (Glyt1), GFAP, Aif1, Gad1/2, and Vglut1/2. Synthetic oligonucleotides in the probes targeted the following mouse mRNAs; Slc6a20 (nt 424–1,313 of Mm-Slc6a20), PTEN (903–1,906 of Mm-Pten-C1, C2), Slc6a9 (nt 958–2,222 of Mm Slc6a9-C2), GFAP (nt 2–1,761 of Mm-GFAP-C3), Aif-1 (nt 31–866 of Mm-Aif-C3), Vglut1 (nt 464–1,415 of Mm-Slc17a7/Vglut1-C2), Vglut2 (nt 1,986–2,998 of Mm-Slc17a6/Vglut2-C3), Gad1 (nt 62–3,113 of Mm-Gad1-C3), and Gad2 (nt 552–1,506 of Mm-Gad2-C2; Advanced Cell Diagnostics, Hayward, CA). The labeled probes were conjugated to Atto 550 (C1), Alexa Fluor 488 (C2), and Atto 647 (C3; Advanced Cell Diagnostics). The labeled probes were conjugated to Atto 550 (C1), Atto 647 (C2), and Alexa Fluor 488 (C3). The sections were hybridized at 40°C with labeled probe mixtures (C1 + C2 + C3) per slide for 2 h. Then, the nonspecifically hybridized probes were removed by washing the sections, three times each in 1× wash buffer at room temperature for 2 min. Amplification steps involved sequential incubations with Amplifier 1-FL for 30 min, Amplifier 2-FL for 15 min, Amplifier 3-FL for 30 min, and Amplifier 4 Alt B-FL at 40°C for 15 min. Each amplifier solution was removed by washing three times with 1× wash buffer for 2 min at RT. Fluorescent images were acquired using TCS SP8 Dichroic/CS (Leica) and analyzed using ImageJ software.

## Total lysates and crude synaptosomes

For whole brain and hippocampus lysates, brain tissues from $Pten^{\Delta C/\Delta C}$ and WT mice were sonicated on ice in RIPA buffer (1% NP40, 1% sodium deoxycholate, 0.1% SDS, 150 mM NaCl, 50 mM Tris–HCl, pH 7.2) containing protease and phosphatase inhibitors, using anvibra-cell ultrasonic liquid processors (sonics and materials inc., CT). Samples were centrifuged at 15,000 *g*, and supernatants were used for protein determination. After immunoblotting, HRP-conjugated and fluorescent secondary antibody signals were detected by Odyssey Fc Dual Mode Imaging System (Li-COR). Signals were quantified using Image Studio Lite (Ver 4.0). The following antibodies were purchased: PTEN (cell signaling, 9559) pPTEN (Ser380/Thr382/383, cell signaling, 9549), AKT (cell signaling, 9272), pAKT (Ser472, cell signaling, 9271), mTOR (cell signaling, 2983), pmTOR (Ser2448, cell signaling, 2971), GluA1 (#1193), GluA2 (#1195), GluA1 S845 (Millipore, 04-1073), GluA1 S831(Millipore, 04-823) and GluN1 (BD bioscince, 556308), GluN2A (millipore, 07-632), GluN2B (Neuromab, 75-101), mGluR1 (Millipore, 07-617), mGluR5 (Millipore, AB5675), GSK3β (cell signaling, 9315), pGSK3β(Ser9, cell signaling, 9336), pERK (Thr202/Tyr204, cell signaling, 9101), ERK (cell signaling, 9102), p38 (cell signaling, 9212), pp38 (Thr180/Tyr182, cell signaling, 9211), β-catenin (BD transduction, 610154), pβCatenin (Ser-552, cell signaling, 9566), pβCatenin (Ser-675, cell signaling, 4176), SLC6A20 (Thermofisher, PA5-68332) for mouse brain lysates, SLC6A20 (Sigma, WH0054716M2) for human neurons, β-actin (Sigma, A5316), and α-tubulin (Sigma, T5168).

## ELISA

Whole-brain tissues were homogenized with ice-cold lysis buffer (50 mM Tris–HCl, pH 7.2, 150 mM NaCl, 1% NP40) containing protease inhibitors. For Slc6a20a and control WT samples, the brains were perfused with PBS before the homogenization. Following brief sonication using an Ultra-Turrax T25 homogenizer, the samples were centrifuged at 15,520 *g* for 20 min at 4°C to pellet the cell debris. The supernatant was collected for direct protein analysis. Tissue concentrations of proline, and glycine were determined using Research ELISA kits for proline (MBS7202081, Mybiosource, CA, USA) and glycine (BA E-2100, ImmuSmol, Talence, France), respectively, following the manufacturer's instructions. Total protein concentration was measured using BCA Protein Assay (Pierce Biotechnology, Rockford, IL, USA). OD was measured using Infinite m200 (Tecan).

## X-gal staining

Mice (P21) were perfused transcardially with 4% paraformaldehyde. Brains were removed and sectioned into 300 μm slices. Slices were incubated in staining solution (5 mM $K_3Fe(CN)_6$, 5 mM $K_4Fe(CN)_6$•$3H_2O$, 2 mM $MgCl_2$, 0.01% deoxycholate, 1 mg/ml X-gal, 0.02% NP-40 in PBS) for 2 h at room temperature. Stained slices were washed four times with PBS and mounted for light microscopy.

## Behavioral assays

All behavioral assays were performed using age-matched male WT mice (2–5 months). All behavior studies were performed during

light-off periods, except for home cage nesting behavior. There were at least 1-day-long rest periods between tests. All data were, unless specified otherwise, analyzed using Ethovision XT 10 (Noldus), and all analyses were performed in a blind manner.

### Laboras test

Long-term (69 h for adult mice and 72 h for juvenile mice) locomotor, climbing, rearing, grooming, eating, and drinking activities were recorded and automatically analyzed using the Laboratory Animal Behavior Observation Registration and Analysis System (LABORAS: Metris). Mice were individually caged in a specialized LABORAS recording environment for the duration of recording and fed *ad libitum*.

### Juvenile play

Direct interaction was performed as previously described (McFarlane *et al*, 2008). Social interaction test sessions were conducted during the first half of the dark cycle in a quiet, dimly lit room illuminated by a single 25 W red light. For juvenile play, P21 mice were brought to the testing room from their home cages for pre-exposure to the experimental conditions. After an hour of isolation, pairs of same sex and genotype but non-sibling mice from different litters were placed in the testing arena, and their interactions were recorded for 30 min. Nose-to-nose sniffing, following, mounting, and allo-grooming were quantified manually as measures of direct social interaction.

### Nesting behaviors

Home cage social behaviors were measured as previously described (Long *et al*, 2004). Mice were individually housed in separate cages with a commercial nesting block. The amount of chewed nestlet and nesting was scored manually every 24 h for 3 days (at 24, 48, and 72 h).

### Maternal homing test

Maternal homing test was performed as previously described (Zhan *et al*, 2014). P19 mice were separated from the mother for at least 30 min before testing. The testing was divided into two stages; (i) nest homing and (ii) maternal homing. (i) For the nest homing, fresh bedding was placed in one corner and bedding from home cage was placed in the opposite corner of a 40 × 40 × 40 cm white acryl box, with the other two corners being empty. The subject was placed in one empty corner, and its movements were recorded for 3 min. (ii) For the maternal homing stage, an empty container and another container containing the mother of the subject were placed in the two previously empty corners. The subject mouse was then placed in the corner with the home cage bedding, and its movements were recorded for 5 min.

### Open field test

Each mouse was placed in the customized open field box (40 cm × 40 cm × 40 cm). The mouse was allowed to move freely inside the box for 1 h, and the activity was video-recorded. The brightness was adjusted to 100 lux for a normal open field test and 0 lux for the light-off version of the open field test. Activities of each mouse such as distance travelled and time spent in the center region of the box were analyzed using EthoVision XT 10 (Noldus).

### Novel object recognition test

For novel object recognition test, each mouse was exposed to two identical objects in the center region of an open field box for 10 min on the first day. Next day, one of the two objects was randomly chosen and switched to a new object, and the mouse was allowed to explore the two objects for 10 min. The 10-min exploration on the second day was used to evaluate novel object exploration. The reaction time toward the object was defined as time a mouse spent sniffing for longer than 1 s. When the mouse touched or climbed over the object, these activities were not counted as reaction time.

### Self-grooming, digging, and marble burying

For home cage self-grooming test, each mouse was placed into a new home cage without bedding and allowed to freely move. Self-grooming activity during 15 min was analyzed. The light in the booth was adjusted to 50 lux. Self-grooming was defined as stroking or scratching of its face or body, or licking its body parts. For digging test, home cages were filled with 2-cm-deep beddings, where mice were placed for 5 min, and the activities were measured and analyzed. Digging was defined as digging out beddings using its head or forelimbs. Self-grooming and digging were scored as the duration of each behavior in a double-blind manner. Marble burying was performed as described previously (Deacon, 2006). Briefly, mice were placed in the corner of a cage, containing 3-cm-thick beddings and 21 marbles (metal, 2 cm in diameter, ordered 3 by 7), Mice were allowed to freely explore the cage for 30 min. Afterward, the number of marbles buried were counted in a blind manner. A marble was considered to be buried when the two-thirds of its height was hidden in beddings.

### Three-chamber test

Three-chamber apparatus has been previously described (Nadler *et al*, 2004; Silverman *et al*, 2010). Mice were isolated in a single cage for 4 days prior to the test, while the stranger mice (129S1/SvlmJ strain) were group-housed (4–6 mice). The test consisted of three phases: empty-empty (habituation), stranger 1-object (S1-O), and stranger 1 (old stranger)-stranger 2 (new stranger; S1–S2) phases. The test was conducted after 30-min habituation in an experimental booth. The white acrylic three-chambered apparatus (60 × 40 × 20 cm) included two small containers for an object or a stranger mouse in the upper or lower corner of the two side chambers. In the first habituation phase, a test mouse was placed in the center area of the three-chambered apparatus and allowed to freely explore the environment for 10 min. In the second S1-O phase, a stranger mouse (S1) and an inanimate blue cylindrical object (O) were placed in the two corner containers. A stranger mouse was randomly positioned in the left or right chamber. The test mouse was allowed to explore the stranger mouse or the object freely. In the third S1–S2 phase, the object was replaced with a new stranger mouse (S2). The test mouse was allowed to freely explore and

interact with both stranger mice. For the analysis, sniffing times were measured using EthoVision XT 10 (Noldus) software. Sniffing was defined as the nose part of the test mouse being positioned within 20% from a container.

## Morris water maze test

This test was performed as described previously (Won *et al*, 2012). A white plastic tank (120 cm) was filled with water with white paint until the platform was hidden under depth of 1 cm of water. Mice were trained to swim to find the hidden platform (10 cm diameter) and stay for 15 s. There were three trials per day. Mice were guided to the platform by human hands if mice failed to find the platform within 60 s. Training continued until the average time for reaching the platform of WT mice fell below 20 s. The probe test was performed once on the test day for 1 min. The reversal memory after the probe test was performed by switching the location of the platform to the opposite quadrant. For long-term memory, mice underwent same training as above, and tested after 7 days. The data were analyzed using EthoVision XT 10 (Noldus).

## Elevated plus-maze and light-dark test

For the elevated plus-maze test, mice were singly placed on the center of the elevated plus maze and allowed to explore for 10 min. The apparatus was made of gray acrylic plates and elevated 50 cm from the floor, with two open arms (30 × 5 × 0.5 cm, 300 lux) and two closed arms (30 × 5 × 30 cm, 30 lux). Time spent in open or closed arms and the frequency of entries to each arm were automatically measured using EthoVision XT 10 (Noldus). For the light-dark test, mice were placed in the light chamber with their heads toward the opposite wall from the dark chamber and allowed to explore the light-dark apparatus (20 × 13 × 20 cm, 300 lux for light chamber, 20 × 13 × 20 cm, 0 lux for dark chamber), which has a 5-cm wide entrance between the two chambers. The latency to enter the dark chamber and time spent in light and dark chambers were analyzed using EthoVision XT 10 (Noldus).

## Drug rescue

For pharmacological rescues of behaviors, D-cycloserine/DCS (Sigma) was dissolved in saline, and mice were injected with DCS (20 mg/kg) and sarcosine (100 mg/kg) intraperitoneally 30 min before experiments. For electrophysiology, DCS and sarcosine were dissolved in aCSF, and slices were bath incubated with DCS (10 μM for LTD, 20 μM for LTP) and sarcosine (750 μM for both LTP and LTD) for at least 30 min before the start of experiments.

## Electron microscopy

Mice were deeply anesthetized with sodium pentobarbital (80 mg/kg, i.p.) and were intracardially perfused with 10 ml of heparinized normal saline, followed by 50 ml of a freshly prepared fixative of 2.5% glutaraldehyde and 1% paraformaldehyde in 0.1 M phosphate buffer (PB, pH 7.4). Hippocampus was removed from the whole brain, postfixed in the same fixative for 2 h, and stored in PB (0.1 M, pH 7.4) overnight at 4 C. Sections were cut transversely on a Vibratome at 70 μm. The sections were osmicated with 1%

osmium tetroxide (in 0.1 MPB) for 1 h, dehydrated in graded alcohols, flat embedded in Durcupan ACM (Fluka), and cured for 48 h at 60°C. Small pieces containing stratum radiatum of the hippocampal CA1 and molecular layer of the hippocampal DG were cut out of the wafers and glued onto the plastic block by cyanoacrylate. Ultrathin sections were cut and mounted on Formvar-coated single slot grids. For quantification of excitatory synapses, brain sections were stained with uranyl acetate and lead citrate, and examined with an electron microscope (Hitachi H-7500; Hitachi) at 80 kV accelerating voltage. Twenty-four micrographs representing 368.9 $\mu m^2$ neuropil regions in each mouse were photomicrographed at a 40,000× and used for quantification. A number of spines (PSD density), the proportion of perforated spines, PSD length, and PSD thickness from three WT and $Pten^{\Delta C/\Delta C}$ mice were quantified. The measurements were all performed by an experimenter blind to the genotype. Digital images were captured with GATAN Digital Micrograph software driving a CCD camera (SC1000 Orius; Gatan) and saved as TIFF files. The brightness and contrast of the images were adjusted in Adobe Photoshop 7.0 (Adobe Systems).

## Antisense ODN (ASO)

Synthesis and purification of all chemically modified ASO was performed as follows (IDT, CA, USA): All antisense oligonucleotides were 25 bp in length, included five 2′-O-methyl-modified nucleotides at each end of the oligonucleotide, with 15 DNA nucleotides in the center, and were phosphorothioate modified in all positions, and all cytosine residues were 5′-methylcytosines, as described previously (Meng *et al*, 2015). The sequences of the ASO was 5′-CACTG AGGCC GTGCC TTCTC CATGT-3′ directed against mouse *Slc6a20a* (NM_139142.2; confirmed unique in mouse with BLASTN), a nucleotide sequence that has been used previously to knockdown *Slc6a20a* expression (Anas *et al*, 2008).

### ICV ASO injection

ASOs were delivered to the mouse brain by intracerebroventricular (ICV) injection, as described previously (Scoles *et al*, 2017). Injection was carried out using a Hamilton syringe (10 μl) and 33-gauge blunt needle. The amount of injection was 2 μl of 50 μg/μl ASO diluted in saline for a total of 100 μg (12 nmol) for 2- to 4-month adult mice. Control mice received the same volume of saline. Injections were made under anesthesia with a mixture of oxygen and isoflurane, using a Kopf stereotaxic frame. For induction, 4% isoflurane anesthesia was used for 5 min, and the concentration was lowered to 2% during injections.Stereotaxic coordinates were (from bregma) −0.46 mm anteroposterior AP, −1.0 mm medio-lateral ML, and −2.3 mm dorsoventral DV for 2–4-month adult mice. Needles were removed 5 min after ASO delivery (0.2 μl/min). After injection, mice were maintained in a clean home cage and warmed with infrared light source while waking up from anesthesia. The WB samples were prepared using mice 3 days after the surgery, while Laboras tests were performed 6 days after surgery. For mice younger than 4 weeks, total of 50 μg of ASO was injected at the coordinate (from bregma) −0.46 mm anteroposterior AP, −1.0 mm medio-lateral ML, and −2.0 mm dorsoventral DV. Electrophysiology tests were conducted 1–3 weeks after the surgery, and qRT–PCR samples were prepared 1–2 weeks after the surgery.

## Proline and glycine transport assay

Proline and glycine transports were measured using automated patch-clamp electrophysiology (Vaidyanathan *et al*, 2014; Yuan *et al*, 2016). HEK293T cells were transfected with constructs for human SLC6A20 variant 1 (RC215764) and variant 2 (RC215810; human, Origene) and mouse SLC6A20A (MR216972) and mouse SLC6A20B (MR209640; mouse, Origene) using lipofectamine 3000 (Invitrogen). The transfected cells were collected and dissociated 24 h after transfection using accutase (Merck). Measurements were performed with the IonFlux HT instrument (Molecular Devices, Sunnyvale, CA), which utilizes microfluidic compound delivery on timescales below 100 ms, facilitating fast activating ligand-gated ion channel measurements. The IonFlux 32 consists of a 96-well microwell plate etched with microfluidic channels at the plate bottom as previously described by previous reports (Golden *et al*, 2011; Spencer *et al*, 2012). Solutions flow over patched cells, giving a superfusion environment for compound exposure. Using this platform, a large number of cells (20 per trap channel) can be held under voltage clamp and exposed to a series of compound concentrations (0, 1, 10, 50, 100, 500, 1,000, 5,000, 10,000 μM) within a short period of time in parallel across a plate. To test NaCl dependence, ChoCl and NaGluconate solutions were used, either as is, or with 5 mM addition of substrate (glycine, proline, sarcosine).

The experiment consisted of four phases: prime, trap, break, and data acquisition. During the preprime step, the IN well was filled with potassium-based intracellular solution. The trap wells were filled with aCSF, and the compound wells had compound solutions. After the preprime step, the IN well was filled with cells suspended in aCSF. Cells are captured from suspension by applying suction to microscopic channels in ensemble recording arrays. In the trap phase, the fluid in the main channel flowed in pulses (no flow for 4.2 s, followed by fluid flowing for 0.8 s), which allowed cells to be trapped. Once the array is fully occupied, the applied suction breaks the cell membranes of captured cells, establishing a whole-cell voltage clamp. The trapped cells were perforated using a rectangular pressure pulse of 4 psi amplitude for 10 s during the break phase. During the data acquisition phase, the main channel and the trap pressures were constant at 0.16 psi and 6 mmHg, respectively. IonFlux software (IonFlux App) was used for data acquisition and exported into Excel (Microsoft) for data analysis. The data were uploaded to GraphPad Prism 7 (GraphPad Software) for visualization, determination of standard deviations, and non-linear regression analysis using $Y = \text{Bottom} + X*(\text{Top-Bottom})/(\text{EC50} + X)$ for concentration-dependent response curves.

Leak compensation in the analyses of auto-patch results was performed by measuring the initial baseline current at the start and subtracting it from subsequent current values in the presence of differing concentrations of glycine or proline. Data were excluded if a recording did not meet the following criteria: (i) if open channel resistances did not drop to approximately 0.5–0.9 MΩ in the priming step, (ii) If seal resistance did not rise to approximately 6–7 MΩ, (iii) If the resistance did not decrease after the "break" phase, and (iv) if the current value for 10,000 μM glycine/proline was not > 10 μM of the same substrate after leak compensation.

To measure transport currents in "single" HEK293T cells and test the possibility of reversal of the currents across holding potentials,

we performed manual/convential patch-clamp (not auto-patch) experiments. Mouse Slc6a20A constructs were used for transient transfection of subconfluent human embryonic kidney cell (HEK293T) using lipofectamine 3000 (Invitrogen), as described above. Electrophysiological recordings were made on the transfected cells grown on coverglass 24 h post-transfection. To achieve whole-cell patch, borosilicate glass pipettes were pulled with a micropipette puller. The resulting recording pipettes (~ 3 MΩ) were filled with K-gluconate intracellular solution of the following composition (in mM): 137 K-gluconate, 5 KCl, 10 HEPES, 0.2 EGTA, 10 Na2-phosphocreatine, 4 Mg-ATP, 0.5 Na-GTP, in pH 7.3 and 295 mOsm. Glycine and proline-induced currents were recorded in a combined voltage ramp / solution exchange protocol. Initially, the HEK cell was patched, and a stable baseline was achieved at holding potential ranging from −90 to +60 mV (in 30 mV increments). Then, a +100 to −100 mV ramp was applied within 1 s before the cell was held again at the initial holding potential. Right after establishing a new voltage, the extracellular solution was changed from no-glycine/proline ACSF to ACSF containing 5 mM glycine/proline. Changes in the current required to hold the cell at the desired membrane potential were interpreted as currents induced by sodium ions entering the cell. After steady-state currents were produced by the perfusion of the recording chamber with glycine/proline-containing extracellular solution, the extracellular solution was once again changed back to ACSF without glycine/proline to observe a gradual decrease in the induced current as substrates were diluted away.

## Human neuron culture

HUES6 obtained from HSCI iPS Core was maintained on MEF (ATCC, #SCRC-1040) in human embryonic stem cell (hESC) medium comprising DMEM/F12 (Gibco #12400024) with MEM NEAA (Gibco #11140050), 14.3 mM sodium bicarbonate (Sigma #S5761), 1 mM L-glutamine (Sigma #S5792), 100μM β-Mercaptoethanol (Merck #M3148), 20% of knockout serum replacement (Gibco #10828028), and 10 ng/ml bFGF (R&D systems #4114-TC) and passed using collagenase IV (Gibco #17104019). To generate floating embryoid bodies (EBs), hESC colonies were dissociated with collagenase IV and plated onto petri dishes in hESC medium. A day after floating culture, to obtain neural progenitor cells (NPCs), the EBs were treated with LDN (Selleckchem #S2618) and SB-431542 (Cayman #CAY-13031) in DMEM/F12 + Glutamax (Gibco #10565042) plus N2 and B27 supplements (Gibco #17502048 and #12587010). The treatment was continued for 7 days, followed by plating onto growth factor reduced matrigel (BD #354230)-coated dishes in DMEM/F12 plus N2 and B27 supplements (N2B27 medium) and 1 μg/ml laminin (Gibco #23017015) to facilitate the attachment of the EBs. Within a few days, rosettes were manually collected and dissociated with Accutase (Innovative Cell Technologies #AT104) and plated onto poly-L-ornithine (Sigma #P3655)/laminin-coated dishes with NPC medium (N2B27 medium plus 20 ng/ml bFGF). To differentiate NPCs into neurons, NPCs were plated into poly-L-ornithine/laminin-coated plates in N2B27 medium in the presence of 100 U/ml penicillin-streptomycin (Gibco #14140122), 200 nM ascorbic acid (Sigma), 500 μg/ml dcAMP (Selleckchem), 20 ng/ml BDNF (PeproTech #450-02), 20 ng/ml GDNF (PeproTech #450-10), and 1 μg/ml laminin as previously described (Brennand *et al*,

**The paper explained**

**Problem**

Glycine transporters play important roles in the regulation of brain levels of glycine, an amino acid acting as a co-agonist of glutamate to stimulate NMDA-type glutamate receptors (NMDARs). Whole-brain glycine levels are tightly regulated by glycine transporters, which transport glycine to several types of brain cells, to maintain optimal levels of NMDAR regulation. Glycine transporters (GlyT1 and GlyT2), well known to regulate brain glycine levels, have been considered to be important therapeutic targets because their inhibition would increase brain glycine levels and NMDAR activity for brain disorders with limited NMDAR function such as schizophrenia. However, it remains unclear whether other amino acid transporters in the brain can also regulate brain glycine levels and NMDAR function.

**Results**

This study finds that SLC6A20A, previously reported to transport mainly proline, can additionally transport glycine in the brain. The unexpected glycine-transporting activity of SLC6A20A was found in mice carrying a mutant form of PTEN, which regulates mTOR signaling. These mice showed abnormally increased SLC6A20A levels, decreased brain glycine levels, and decreased NMDAR function, which are all casually associated. Conversely, SLC6A20A-mutant mice show increased brain glycine levels and NMDAR function. Lastly, both mouse and human SLC6A20A proteins mediate glycine and proline transports at similar levels.

**Impact**

These results suggest that SLC6A20 is a novel regulator of glycine and proline levels in the brain. In addition, inhibition of SLC6A20 activity by pharmacological or molecular biological methods and consequent increases in brain glycine levels and NMDAR activity could be a potential strategy for the treatment of brain disorders involving NMDAR hypofunction such as schizophrenia and autism spectrum disorders.

2011). Protocols describing the use of human ESCs were approved in accordance with the ethical requirements and regulations of the Institutional Review Board of KAIST (IRB #KA2018-61, KH2020-55).

**Data acquisition statistical analysis**

We performed our behavioral analysis in a double-blind manner. All data were expressed as mean values with standard error of mean (SEM). All statistical analyses were performed using GraphPad Prism software (version 7.0). Statistical details are summarized in Appendix Table S1.

# Data availability

The datasets produced in this study are available in the following databases:

RNA-Seq data: Gene Expression Omnibus GSE119236 (https://www.ncbi.nlm.nih.gov/geo/query/acc.cgi?acc=GSE119236).

**Expanded View** for this article is available online.

## Acknowledgements

This work was supported by the National Research Foundation of Korea (NRF-2017R1A5A2015391 to Y.C.B.; NRF-2017R1A2B3002862 to J.W.K.; NRF-2019R1C1C1010482 to J.H.; NRF-2017M3C7A1079692 to H.Kim), the Korea Institute of Science and Technology Information (K-18-L12-C08-S01 to H.Kang), and the Institute for Basic Science (IBS-R002-A1 to J.H. and IBS-R002-D1 to E.K.).

## Author contributions

MB, JDR, RK, TY, and HO performed behavioral experiments. MB and HJ performed immunoblot and immunohistochemistry experiments. MB, JDR, RK, HK and DoK performed electrophysiological experiments. MB and SH performed ChIP experiments. MB and YK maintained Pten-mutant mice. HKang, HJ and MB performed RNA-Seq experiments. HMH performed electron microscopy experiments. EY and JYK performed FISH experiments. SL and DaK performed human neuron experiments. SSK performed mass spectrometry experiments. MB and JDR performed SLC6A20A transport experiments. JH, YCB, HKim, SA, AMC, DL, JWK, and EK designed the experiments and wrote the manuscript.

## Conflict of interest

The authors declare that they have no conflict of interest.

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
