## [Review Process File · EMBO Molecular Medicine]

SLC6A20 transporter: a novel regulator of brain glycine homeostasis and NMDAR function

Mihyun Bae, Junyeop Roh, Youjung Kim, Seong Soon Kim, Hye Min Han, Ester Yang, Hyojin Kang, Suho Lee, Jin Yong Kim, Ryeonghwa Kang, Hwajin Jung, Taesun Yoo, Hyosang Kim, Doyoun Kim, Heejeong Oh, Sungwook Han, Dayeon Kim, Jinju Han, Yong Chul Bae, HYUN Kim, Sunjoo Ahn, Andrew Chan, Daeyoup Lee, Jin Woo Kim, and Eunjoon Kim

DOI: [10.15252/emmm.202012632](https://doi.org/10.15252/emmm.202012632)

Corresponding author(s): Eunjoon Kim (kime@kaist.ac.kr)

Review Timeline:

Submission Date:	29th Apr 20
Editorial Decision:	4th Jun 20
Revision Received:	2nd Oct 20
Editorial Decision:	20th Oct 20
Revision Received:	22nd Oct 20
Accepted:	19th Nov 20

Editor: Celine Carret & Jingyi Hou

Transaction Report:

Thank you for the submission of your revised manuscript to EMBO Molecular Medicine. We have now heard back from three referees of the four who have initially evaluated your study.

You will see that while referees #1 and #2 are globally satisfied and only request minor revisions, referee #3 is not satisfied and still raised important concerns.

We have looked attentively into these concerns and would like to propose the following plan to you:

Regarding point 1 of ref. #3, in vitro transport of Glycine.

Please provide actual traces for this experiment (fig. 7), include the needed controls and detail the patch-clamp calibration in order to answer to referee 3's concerns.

Regarding point 2:

- FISH: please repeat the experiment on SLC6A20^{-/-} mouse brain
- clarify and explain the high values of glycine measured
- no action required for the micro dialysis experiment

Regarding point 3: Please explain better what you meant and exert caution in the interpretation of your data

Please make sure to provide a point-by-point letter carefully answering all issues raised by the three referees and making the appropriate changes in your manuscript, experimentally as indicated above. Please note that this is the last invitation to revise your article, therefore I would encourage you to follow our indications as much as possible. Depending on the nature of the revisions, this may be sent back to the referees for another round of review

Please re-submit your article as soon as ready. If you anticipate that it may take more than 3 months, I would appreciate a letter of update.

In order to gain time, should the manuscript move forward, please also address the below editorial recommendations for formatting of your revised article.

I look forward to seeing a revised form of your manuscript as soon as possible.

***** Reviewer's comments *****

Referee #1 (Comments on Novelty/Model System for Author):

Authors have used PTEN mutant mice and SLC6A20A knockout mice to prove that SLC6A20 can function as a glycine transporter.

Referee #1 (Remarks for Author):

The revised manuscript by Bae et al. has provided more evidence showing that SLC6A20, which is upregulated in mice carrying a mutant PTEN protein, functions as a glycine transporter that decreases brain glycine levels and NMDAR function. It suggests a potential therapeutic avenue to target SLC6A20 for brain disorders that involve NMDAR hypofunction. Authors have painstakingly addressed prior concerns raised by 4 reviewers, and newly added data have significantly improved the rigor of the conclusion. I have a few minor concerns that need to be addressed.

1. Regarding the direct measurement of NMDAR function, authors have added NMDAR mEPSC data (Fig. 1H). While the statistics shows a significant reduction of NMDAR mEPSC amplitude in PTEN mutant mice, the representative traces do not have clear differences. More representative examples should be used.

2. Authors stated that they have used a "modified antisense oligonucleotides with enhanced cell permeability" (p18). It is unclear what modification was used to increase cell permeability.

3. In GSEA analysis, why "5917 gene sets in the C5 (gene ontology) category" were used?

4. Authors have linked the altered repetitive climbing behavior to SLC6A20 dysregulation of NMDARs. It will be interesting to know whether SLC6A20A knockout mice also have changed climbing behavior.

5. The newly generated mouse line lacks exon 3 of the Slc6a20a gene. Why was only exon 3 deleted? Fig. 6A should be moved to supplementary figures, and key information about the generation and characterization of Slc6a20a-mutant mice (Fig. EV10) could be moved to Fig. 6A.

6. Fig. 7B has very weak signals (v1 and v2) in human neurons. The blots for human neurons need to be intensified, so the signals are visible. Why are there two blots (left) above actin? What is "2 wks unboiled"? What about other time points unboiled? The description about Fig. 7B is too brief and very unclear.

7. Page 3, "...mouse line that lacks the C-terminal tail of expressing PTEN lacking of the PTEN protein" needs to be fixed.

8. Page 4, "(LTD)26, 34" needs to change the reference citation format.

9. Page 5, "repetitive behavior" is better changed to be "repetitive climbing behavior"

Referee #3 (Comments on Novelty/Model System for Author):

See below, but the electrophysiological determination of SLC6A20 substrate specificity for glycine is likely to be an artifact without the appropriate controls.

Referee #3 (Remarks for Author):

The authors provide a large set of new data in their revised manuscript, with the aim to bold their interpretation that the deletion of the PDZ-binding domain of PTEN in a mouse model leads to a transcriptional increase in *slc6a20a*, a proline-specific transporter expressed primarily in the meninges and choroid plexus that would deplete glycine and suppress two forms of NMDA-dependent plasticity, with the consequence of increasing the climbing behavior of juvenile mice. Although I appreciated the effort made by the authors, with the addition of new experiments, the inclusion of some raw data, controls, and statistics, I am still unconvinced by the author confusing interpretation that *slc6a20a* might play a physiological role in regulating NMDAR at excitatory synapses.

Point 1

Proving that SLC6A20 transports glycine is undoubtedly a pivotal element for the author resubmission as it is not treated as a minor point in the supplementary material but as the final figure of this complex paper. Using an automated patch clamp system, the authors measured nearly-identical glycine and proline evoked currents in HEK cells transfected with different SLC6A20 isoforms, and jump directly to the conclusion that SLC6A20A mediates "proline and glycine transport equally efficiently", with submicromolar EC50 for the human isoform. However, the authors cannot eliminate so easily the discrepancy with the proline-specificity of SLC6A20 that has been done in *Xenopus* oocytes (Bröer S et al., 2008; Kowalczyk S, 2005; Takanaga H et al., 2005), by merely invoking the exquisite "sensitivity" of the patch-clamp technique or post-translational differences or cellular environment that would magically change the substrate-site, which is deeply buried in the transporter structure, inside the membrane.

The argument of the higher sensitivity of the patch-clamp technique is meaningless for this purpose because the elementary current of ion-coupled transporters, which is typically in the 10-19 - 10-18A range, is about a million-fold lower than the pA resolution of the patch-clamp technique. If there is an interest in the patch-clamp technique, it is for controlling the composition of the intracellular compartment (not used here) and to achieve much faster voltage clamp (not used here). So, a transporter current recorded in HEK cells is not different from the one recorded in *Xenopus* oocytes and requires high rate of protein synthesis and membrane trafficking. At this

game, *Xenopus* oocytes have a major advantage because of their large membrane excess and built-in protein synthesis factory that can efficiently synthesize several billion (10¹⁰-10¹²) transporters within few days post cRNA injection. In contrast, mammalian cells may have difficulty for trafficking and maintaining millions of proteins at their membrane for electrophysiology. They are suitable for uptake experiments because of the higher sensitivity of the labeling technique that integrates uptake over time and requires less transporter expression. For example, the fuzzy expression of SLC6A20 shown in figure 7A is far from ideal membrane localization for electrophysiological recordings.

The problem with the data, as presented by the authors, is that they do not show the actual current traces but only a summary plot of current amplitudes. So, it is not possible to examine what has been "carefully" measured, an adverb not often used for analyzing transporter currents in *Xenopus* oocytes because the amplitude is steady and robust. Brief (100 ms) applications of glycine or proline over this broad range of concentrations (1 μ M-10mM) should have produced major changes in kinetic profiles toward the steady-state transport current, that would have reassured about the origin of the measured current. Otherwise, it is tempting to think that a current artifact has been measured. An automated patch clamp setup developed for massive pharmacological screening, with loose seal and multiple cells in each well, may be prone to small leakage current artifacts due to rapid change in solutions that would only be marginal when studying large conductance channels, but a problem for measuring small transporter currents. This artifact hypothesis would explain why both mouse isoforms generate the same current while SLC6A20B is not electrogenic in *Xenopus* oocytes (Kowalczyk S et al., 2005), and also why the current amplitude of the two human isoforms are almost constant over the entire range of proline and glycine concentrations. Fast turnover and submicromolar EC₅₀ do not match together.

To show that the recorded current is a glycine-coupled Na⁺ current and to exclude an artifact origin, the authors should have shown the actual traces and performed the following controls: 1) the glycine and proline evoked currents should be Na⁺ and Cl⁻ dependent (i.e., no current evoked by 10 mM amino acid in choline Cl and NaGluconate solution); 2) the evoked current should not have a reversal potential, which is a key signature for tightly coupled Na⁺-transporters; 3) the current should not be activated by non-transported substrates, but should transport sarcosine, for example. If these three points were confirmed, the authors would gain confidence in showing glycine uptake directly in HEK cells expressing SLC6A20A, allowing them to revise the substrate specificity of SLC6A20A. Without these controls, figure 7 does not provide convincing evidence supporting the author's central hypothesis, and SLC6A20a should still be considered as a proline-specific transporter.

Point 2

The in-situ hybridizations confirm that SLC6A20a is expressed mostly in the meninges and choroid plexus of the mouse brain, which agrees with previous transcriptomic analysis that showed high expression in endothelial cells and notably the pericytes (He L et al., 2016) around capillaries, but with low expression or no detection in whole-brain mouse, in contrast to GlyT1 (Dahlin A et al., 2009 ; Dörrbaum AR et al., 2018; Fornasiero EF et al., 2018; Kadakkuzha BM et al., 2015). The sparse labeling found here in the hippocampus and cortex is puzzling because SLC6a20a is expected to be present on all capillary walls (endothelial/pericyte). A control experiment with the SLC6a20a^{-/-} mice would have excluded non-specific labeling.

How the lack of the PDZ-domain of PTEN at excitatory synapses increases the transcription of slc6a20a in the meninges, choroid plexus, and capillaries? In any case, the synaptic cartoon in figure 5 is misleading (a glycine transporter, presumably SLC6A20 but unlabeled is drawn on the microglia) since SLC6A20 is not a synaptic protein.

Figure 3A shows a 50000 ratio between glycine (2 mg) and proline (40 ng) values in the whole brain of mice. This ratio is not reasonable, and confirm that these ELISA tests that have been developed for urinary samples are not reliable for the quantification of whole-brain amino acids. Furthermore, glycine values for the WT group differ STRONGLY between figure 3A (1.951 mg/mg protein) and figure 5D (0.672 mg/mg protein), which is even below the value for the DC group (0.896 mg/mg protein) in figure 3A. Finally, it is surprising that such an extraordinary value for glycine (2 mg /mg protein) did not catch the attention of the authors that something was likely to be wrong. Assuming a yield of protein extraction of 100 µg protein/mg weight and an average adult brain volume of 0.5 ml (500 mg weight), the glycine content would be 2.7 M for the whole brain!

In contrast, the microdialysis experiments are self-coherent as glycine and proline values are in a similar concentration range, and the values in control sound reasonable, although low compared to other reports of microdialysis. However, with such complex methodology, with no control and a single measurement per mice, these data are not enough to believe that 1.3 fold increase of a low abundant proline transporter located around capillaries and meninges could extrude 2/3 of glycine from the brain, reaching basal concentration so low that no amino acid transporters would have the driving force to support such enormous gradient and would reverse and release.

Point 3

The authors confirm "that the PTEN PB motif is required for synaptic translocation of PTEN during LTD" (Jurado S et al., 2010), but not for LTP induced by theta-burst stimulation (Knafo S et al., 2016). In contrast, they found that high-frequency tetanic stimulation failed to produce the late phase of LTP in PTEN DC/DC mice. Because LTD and LTP-HFS are NMDA-dependent, the authors conclude that NMDA signaling is suppressed in this mutant mouse and then examine NMDA current and the glycine depletion hypothesis. However, theta-burst LTP, which is a more robust experimental paradigm to trigger stable LTP than high-frequency tetanic stimulation (Hoffman DA et al., 2002; Larson J and Munkácsy E, 2015), is also dependent on NMDAR activation (Larson J and Munkácsy E, 2015) and it is not clear why the authors bypass this fact that contradicts their interpretation.

Kim's group already described NMDA-dependent suppression of LTD and LTP in the analysis of the Shank-/- knockout mice (Won H et al., 2012). Like here, the hypofunction of NMDA receptors was the immediate cause since the plasticity was restored by adding D-Cycloserine (Won H et al., 2012), although Shank inactivation is unlikely to alter the extracellular glycine level. Therefore, interfering with the PSD-95 may lower NMDA signaling by many mechanisms. For example, we can speculate that changing the distribution of synaptic vs. extrasynaptic NMDA receptors would shift the coagonist sensitivity of NMDA receptor from D-serine (synaptic) to glycine (extrasynaptic) (Papouin T et al., 2012), and thus provide similar change in plasticity.

Selection of minor problems.

It is hard to accept that a 25% reduction in the number of excitatory synapses does not affect the input/output relationship. What does mean synapses survival for fiber recruitment?

The repeated usage (35) of the adjective strong, strongly, stronger to describe each results is often not supported by the actual data.

Page 8: "strong decrease" is 25% reduction of synaptic element without any consequence

Page 8: LTP was "strongly suppressed": suppressed or strongly reduced would make more sense.

Page 8: again, LTD was "strongly suppressed."

Page 9: NMDA was "strongly reduced"

Page 11 "the strongly increased expression of Slc6a20a": Here, strongly refers to a 1.3 fold overexpression, which seems marginal. Seven occurrences of strong are found in page 11.

Page 12 "deletion of the PTEN PB motif induces a strong increase in transmembrane transport-related functions".

Page 12 "scl6a20a gene showed a strong increase in the expression"

Page 12 "exhibiting fairly strong expression in astrocyte".

etc

Bröer S, Bailey CG, Kowalczyk S, Ng C, Vanslambrouck JM, Rodgers H, Auray-Blais C, Cavanaugh JA, et al. (2008), Iminoglycinuria and hyperglycinuria are discrete human phenotypes resulting from complex mutations in proline and glycine transporters. *J Clin Invest* 118:3881-3892.

Dahlin A, Royall J, Hohmann JG, Wang J (2009), Expression Profiling of the Solute Carrier Gene Family in the Mouse Brain. *J Pharmacol Exp Ther* 329:558-570.

Dörrbaum AR, Kochen L, Langer JD, Schuman EM (2018), Local and global influences on protein turnover in neurons and glia. *Elife* 7:e34202.

Fornasiero EF, Mandad S, Wildhagen H, Alevra M, Rammner B, Keihani S, Opazo F, Urban I, et al. (2018), Precisely measured protein lifetimes in the mouse brain reveal differences across tissues and subcellular fractions. *Nat Commun* 9:4230.

He L, Vanlandewijck M, Raschperger E, Mäe MA, Jung B, Lebouvier T, Ando K, Hofmann J, et al. (2016), Analysis of the brain mural cell transcriptome. *Sci Rep-uk* 6:35108.

Hoffman DA, Sprengel R, Sakmann B (2002), Molecular dissection of hippocampal theta-burst pairing potentiation. *Proc National Acad Sci* 99:7740-7745.

Jurado S, Benoist M, Lario A, Knafo S, Petrok CN, Esteban JA (2010), PTEN is recruited to the postsynaptic terminal for NMDA receptor-dependent long-term depression. *The EMBO Journal* 29:2827-2840.

Kadakkuzha BM, Liu X-A, McCrate J, Shankar G, Rizzo V, Afinogenova A, Young B, Fallahi M, et al. (2015), Transcriptome analyses of adult mouse brain reveal enrichment of lncRNAs in specific brain regions and neuronal populations. *Front Cell Neurosci* 9:63.

Knafo S, Sánchez-Puelles C, Palomer E, Delgado I, Draffin JE, Mingo J, Wahle T, Kaleka K, et al. (2016), PTEN recruitment controls synaptic and cognitive function in Alzheimer's models. *Nat Neurosci* 19:443-453.

Kowalczyk S, Bröer A, Munzinger M, Tietze N, Klingel K, Bröer S (2005), Molecular cloning of the mouse IMINO system: an Na⁺- and Cl⁻-dependent proline transporter. *Biochemical Journal* 386:417-422.

Larson J, Munkácsy E (2015), Theta-burst LTP. *Brain Res* 1621:38-50.

Papouin T, Ladépêche L, Ruel J, Sacchi S, Labasque M, Hanini M, Groc L, Pollegioni L, et al. (2012), Synaptic and extrasynaptic NMDA receptors are gated by different endogenous coagonists. *Cell* 150:633-646.

Takanaga H, Mackenzie B, Suzuki Y, Hediger MA (2005), Identification of Mammalian Proline Transporter SIT1 (SLC6A20) with Characteristics of Classical System Imino. *Journal of Biological Chemistry* 280:8974-8984.

Won H, Lee H-R, Gee HY, Mah W, Kim J-I, Lee J, Ha S, Chung C, et al. (2012), Autistic-like social behaviour in Shank2-mutant mice improved by restoring NMDA receptor function. *Nature* 486:261-265.

Point-by-point responses to reviewers' comments**Re: SLC6A20 transporter: a novel regulator of brain glycine homeostasis and NMDAR function**

By Bae, Roh, Kim, Kim, Han, Yang, Kang, Lee, Kim, Kang, Jung, Yoo, Kim, Kim, Oh, Han, Kim, Han, Bae, Kim, Ahn, Chan, Lee, Lee, Kim, and Kim

******* Reviewer's comments *********Referee #1 (Comments on Novelty/Model System for Author):**

Authors have used PTEN mutant mice and SLC6A20A knockout mice to prove that SLC6A20 can function as a glycine transporter.

Referee #1 (Remarks for Author):

The revised manuscript by Bae et al. has provided more evidence showing that SLC6A20, which is upregulated in mice carrying a mutant PTEN protein, functions as a glycine transporter that decreases brain glycine levels and NMDAR function. It suggests a potential therapeutic avenue to target SLC6A20 for brain disorders that involve NMDAR hypofunction. Authors have painstakingly addressed prior concerns raised by 4 reviewers, and newly added data have significantly improved the rigor of the conclusion. I have a few minor concerns that need to be addressed.

→ We appreciate the encouraging remarks of the reviewer.

1. Regarding the direct measurement of NMDAR function, authors have added NMDAR mEPSC data (Fig. 1H). While the statistics shows a significant reduction of NMDAR mEPSC amplitude in PTEN mutant mice, the representative traces do not have clear differences. More representative examples should be used.

→ We appreciate the careful comment. We replaced the current traces with better ones.

2. Authors stated that they have used a "modified antisense oligonucleotides with enhanced cell permeability" (p18). It is unclear what modification was used to increase cell permeability.

→ We used 2'-O-methyl (2'-OMe) modification; we clarified it in Methods.

3. In GSEA analysis, why "5917 gene sets in the C5 (gene ontology) category" were used?

→ That was the total number of gene sets in the C5 category (https://www.gsea-msigdb.org/gsea/msigdb/collection_details.jsp#C5). We changed the text as follows to minimize confusion: "5917 gene sets" was changed to "the gene sets".

4. Authors have linked the altered repetitive climbing behavior to SLC6A20 dysregulation of NMDARs. It will be interesting to know whether SLC6A20A knockout mice also have changed climbing behavior.

→ *Slc6a20a*^{-/-} mice display normal repetitive climbing likely because *Slc6a20a* up- and down-regulations have distinct impacts on mouse behaviors. We plan to publish this result and other electrophysiological and behavioral phenotypes of *Slc6a20a*^{-/-} mice in future studies; reviewer's understanding would be much appreciated.

5. The newly generated mouse line lacks exon 3 of the *Slc6a20a* gene. Why was only exon 3 deleted? Fig. 6A should be moved to supplementary figures, and key information about the generation and characterization of *Slc6a20a*-mutant mice (Fig. EV10) could be moved to Fig. 6A.

→ The mouse line was obtained from EMMA (EMMA ID: 09248, *Slc6a20a tm1b(KOMP)Wtsi*). We thus did not have a control over the exon to be deleted. However, we confirmed that this deletion decreases *Slc6a20a* mRNA and protein levels (Fig. 6B,C).

We agree with the reviewer and switched the two figures; now Fig. 6A shows KO strategy, and Fig. Ev10A shows gross brain morphology.

6. Fig. 7B has very weak signals (v1 and v2) in human neurons. The blots for human neurons need to be intensified, so the signals are visible. Why are there two blots (left) above actin? What is "2 wks unboiled"? What about other time points unboiled? The description about Fig. 7B is too brief and very unclear.

We intensified the weak signals (v1 and v2), as requested.

The multiple blots above the actin blot are identical blots with different signal intensities (by short, long, and longer exposures); we clarified this in the figure panel.

Boiled membrane proteins usually aggregate and get stuck at the start location of Western blot gels. An increase in the *Slc6A20* band signal (2-week time point) by unboiling suggests that the band is a membrane protein. We did not try to unboil other samples at other time points because one sample would be enough for proof of concept. We clarified these points in the figure legend.

7. Page 3, "...mouse line that lacks the C-terminal tail of expressing PTEN lacking of the PTEN protein" needs to be fixed.

→ Thanks, we corrected it.

8. Page 4, "(LTD)26, 34" needs to change the reference citation format.

→ Thanks, we corrected it.

9. Page 5, "repetitive behavior" is better changed to be "repetitive climbing behavior"

→ Thanks, we corrected it.

Referee #3 (Comments on Novelty/Model System for Author):

See below, but the electrophysiological determination of SLC6A20 substrate specificity for glycine is likely to be an artifact without the appropriate controls.

Referee #3 (Remarks for Author):

The authors provide a large set of new data in their revised manuscript, with the aim to bold their interpretation that the deletion of the PDZ-binding domain of PTEN in a mouse model leads to a transcriptional increase in *slc6a20a*, a proline-specific transporter expressed primarily in the meninges and choroid plexus that would deplete glycine and suppress two forms of NMDA-dependent plasticity, with the consequence of increasing the climbing behavior of juvenile mice.

Although I appreciated the effort made by the authors, with the addition of new experiments, the inclusion of some raw data, controls, and statistics, I am still unconvinced by the author confusing interpretation that *slc6a20a* might play a physiological role in regulating NMDAR at excitatory synapses.

→ We now provide additional datasets to support that SLC6A20A transports glycine in addition to proline.

Point 1

Proving that SLC6A20 transports glycine is undoubtedly a pivotal element for the author resubmission as it is not treated as a minor point in the supplementary material but as the final figure of this complex paper. Using an automated patch clamp system, the authors measured nearly-identical glycine and proline evoked currents in HEK cells transfected with different SLC6A20 isoforms, and jump directly to the conclusion that SLC6A20A mediates "proline and glycine transport equally efficiently", with submicromolar EC50 for the human isoform. However, the authors cannot eliminate so easily the discrepancy with the proline-specificity of SLC6A20 that has been done in *Xenopus oocytes* (Bröer S et al., 2008; Kowalczyk S, 2005; Takanaga H et al., 2005), by merely invoking the exquisite "sensitivity" of the patch-clamp technique or post-translational differences or cellular environment that would magically change the substrate-site, which is deeply buried in the transporter structure, inside the membrane.

The argument of the higher sensitivity of the patch-clamp technique is meaningless for this purpose because the elementary current of ion-coupled transporters, which is typically in the 10⁻¹⁹ - 10⁻¹⁸A range, is about a million-fold lower than the pA resolution of the patch-clamp technique. If there is an interest in the patch-clamp technique, it is for controlling the composition of the intracellular compartment (not used here) and to achieve much faster voltage clamp (not used here). So, a transporter current recorded in HEK cells is not different from the one recorded in *Xenopus oocytes* and requires high rate of protein synthesis and membrane trafficking.

At this stage, *Xenopus* oocytes have a major advantage because of their large membrane excess and built-in protein synthesis factory that can efficiently synthesize several billion (10¹⁰-10¹²) transporters within few days post cRNA injection. In contrast, mammalian cells may have difficulty for trafficking and maintaining millions of proteins at their membrane for electrophysiology. They are suitable for uptake experiments because of the higher sensitivity of the labeling technique that integrates uptake over time and requires less transporter expression. For example, the fuzzy expression of SLC6A20 shown in figure 7A is far from ideal membrane localization for electrophysiological recordings.

The problem with the data, as presented by the authors, is that they do not show the actual current traces but only a summary plot of current amplitudes. So, it is not possible to examine what has been "carefully" measured, an adverb not often used for analyzing transporter currents in *Xenopus* oocytes because the amplitude is steady and robust. Brief (100 ms) applications of glycine or proline over this broad range of concentrations (1 μ M-10 mM) should have produced major changes in kinetic profiles toward the steady-state transport current, that would have reassured about the origin of the measured current. Otherwise, it is tempting to think that a current artifact has been measured. An automated patch clamp setup developed for massive pharmacological screening, with loose seal and multiple cells in each well, may be prone to small leakage current artifacts due to rapid change in solutions that would only be marginal when studying large conductance channels, but a problem for measuring small transporter currents. This artifact hypothesis would explain why both mouse isoforms generate the same current while SLC6A20B is not electrogenic in *Xenopus* oocytes (Kowalczyk S et al., 2005), and also why the current amplitude of the two human isoforms are almost constant over the entire range of proline and glycine concentrations. Fast turnover and submicromolar EC₅₀ do not match together.

To show that the recorded current is a glycine-coupled Na⁺ current and to exclude an artifact origin, the authors should have shown the actual traces and performed the following controls: 1) the glycine and proline evoked currents should be Na⁺ and Cl⁻ dependent (i.e., no current evoked by 10 mM amino acid in choline Cl and NaGluconate solution); 2) the evoked current should not have a reversal potential, which is a key signature for tightly coupled Na⁺-transporters; 3) the current should not be activated by non-transported substrates, but should transport sarcosine, for example. If these three points were confirmed, the authors would gain confidence in showing glycine uptake directly in HEK cells expressing SLC6A20A, allowing them to revise the substrate specificity of SLC6A20A. Without these controls, figure 7 does not provide convincing evidence supporting the author's central hypothesis, and SLC6A20a should still be considered as a proline-specific transporter.

→ We appreciate additional and detailed comments of the reviewer, which were very helpful in improving the manuscript in the following ways.

1) We now show examples of actual traces of glycine/proline-induced currents from automatic patch (auto-patch) experiments with details described in the figure legend (**Fig. EV12**).

2) We additionally measured glycine/proline-induced currents from 'single' HEK293T cells by 'conventional' patch-clamp experiments (not auto-patch experiments) that we routinely use in the laboratory, where glycine/proline elicits similar levels of currents (**Fig. 7C**), similar to the results from auto-patch experiments.

3) We also used this new experimental format (single HEK293T cell patch-clamp experiments) to demonstrate that glycine/proline currents do not have a reversal potential (a key feature of transporters that the reviewer pointed out) (**Fig. 7C**).

4) We now demonstrate, by auto-patch experiments, that glycine/proline-evoked currents of SLC6A20A are NaCl-dependent. Specifically, choline chloride and sodium gluconate replacing Na⁺ and Cl⁻ ions in NaCl, respectively, markedly suppress glycine/proline currents (**Fig. 7B**).

5) We also demonstrate, by auto-patch experiments, that SLC6A20a transports sarcosine (a positive control transport substrate suggested by the reviewer) in a NaCl-dependent manner (**Fig. EV13A**), but not histidine or GABA (negative control substrates) (**Fig. EV13B**).

6) The similar levels of glycine/proline-evoked currents across a range of glycine/proline concentrations were partly because we did not subtract the baseline currents in the absence of glycine/proline (although relatively small) from the currents in the presence of glycine/proline. We corrected it. These improvements in experiments/analyses increased the K_{0.5} values of glycine/proline currents (**Fig. 7A**), which are closer to those observed in *Xenopus* oocytes. We also detailed in Methods how the auto-patch system excludes unqualified currents.

7) We found in the literature that the substrate specificity of a transporter can be changed depending on the cell types used to express and characterize the transporters (Palacin et al, 1998); i.e., GAT-3 can transport both GABA and β-alanine efficiently (~K_m of 18 and 28 μM) in *Xenopus* oocytes (Liu et al, 1993), whereas the same GAT-3 can transport GABA but not β-alanine in HeLa cells (Clark et al, 1992). Yet, GAT-3 can transport β-alanine and be inhibited of GABA transport by β-alanine when stably expressed in LLC-PK₁ cells, a mammalian cell type (Clark & Amara, 1994). We mentioned these results in Discussion.

8) Lastly, many review papers on the characterization of ion channels and transporters in heterologous cell systems describe both the advantages and disadvantages of *Xenopus* oocyte and mammalian cell systems (Clare & Trezise, 2006). A key advantage of mammalian cell systems is that they better mimic the original cell types where channels/transporters are expressed. In addition, we have to point out that *SLC6A20* mutations in humans leads to complex digenic iminoglycinuria (a renal disorder with impaired reabsorption of glycine and imino acids [proline and hydroxyproline]) and hyperglycinuria (Broer et al, 2008), as stated in Discussion, suggesting that *SLC6A20* transports glycine in addition to proline.

Point 2

The in-situ hybridizations confirm that *Slc6a20a* is expressed mostly in the meninges and choroid plexus of the mouse brain, which agrees with previous transcriptomic analysis that showed high expression in endothelial cells and notably the pericytes (He L et al., 2016) around capillaries, but with low expression or no detection in whole-brain mouse, in contrast to GlyT1 (Dahlin A et al., 2009 ; Dörrbaum AR et al., 2018; Fornasiero EF et al., 2018; Kadakkuzha BM et al., 2015). The sparse labeling found here in the hippocampus and cortex is puzzling because SLC6A20a is expected to be present on all capillary walls (endothelial/pericyte). A control experiment with the *Slc6a20a*^{-/-} mice would have excluded non-specific labeling.

→ We performed the suggested control experiments and now show that *Slc6a20a* mRNAs signals are greatly reduced in the *Slc6a20a*^{-/-} mice, in areas including not only the meninges and choroid plexus but also the cortex and hippocampus (**Fig. EV5C,D**).

We also added the following description to Results to highlight the reported *Slc6a20a* expression in brain capillaries (we appreciate this information): “Notably, the meningeal and choroid plexus localization of *Slc6a20* mRNAs agreed with the reported enrichment of *Slc6a20a* expression in pericytes around capillaries, known to regulate BBB functions (Hall et al, 2014; He et al, 2016; Vanlandewijck et al, 2018).”

How the lack of the PDZ-domain of PTEN at excitatory synapses increases the transcription of *slc6a20a* in the meninges, choroid plexus, and capillaries?

→ PTEN is widely expressed in various brain cell types, as supported by our FISH results (**Fig. EV6**). It is therefore possible that Pten C-terminal deletion in the cells of the meninges, choroid plexus, and capillaries may cause cell-autonomous *Slc6a20a* upregulation through nuclear β -catenin translocation that we mentioned in the main text; we clarified this point in Discussion.

In any case, the synaptic cartoon in figure 5 is misleading (a glycine transporter, presumably SLC6A20 but unlabeled is drawn on the microglia) since SLC6A20 is not a synaptic protein.

→ We corrected the cartoon in **Fig. 4A,E** by moving the transporter symbols away from the synapse to the other corner of the microglial cell, which expresses *Slc6a20a*, as shown by our FISH results.

Figure 3A shows a 50000 ratio between glycine (2 mg) and proline (40 ng) values in the whole brain of mice. This ratio is not reasonable, and confirm that these ELISA tests that have been developed for urinary samples are not reliable for the quantification of whole-brain amino acids. Furthermore, glycine values for the WT group differ STRONGLY between figure 3A (1.951 mg/mg protein) and figure 5D (0.672 mg/mg protein), which is even below the value for the DC group (0.896 mg/mg protein) in figure 3A. Finally, it is surprising that such an extraordinary value for glycine (2 mg /mg protein) did not catch the attention of the authors that something was likely to be wrong. Assuming a yield of protein extraction of 100 μ g protein/mg weight and an average adult

brain volume of 0.5 ml (500 mg weight), the glycine content would be 2.7 M for the whole brain!

In contrast, the microdialysis experiments are self-coherent as glycine and proline values are in a similar concentration range, and the values in control sound reasonable, although low compared to other reports of microdialysis. However, with such complex methodology, with no control and a single measurement per mice, these data are not enough to believe that 1.3 fold increase of a low abundant proline transporter located around capillaries and meninges could extrude 2/3 of glycine from the brain, reaching basal concentration so low that no amino acid transporters would have the driving force to support such enormous gradient and would reverse and release.

→ We appreciate these careful comments, which correctly pointed out our embarrassing mistakes in the calculation; our sincere apologies! We now show correct glycine concentrations in **Fig. 3A**, where proline and glycine concentrations are largely in comparable ranges. We also show correct glycine concentrations in **Fig. 5D**, which are ~4 times lower than those in Fig. 3A. This could be attributable to different mouse ages (P21 in Fig. 3A vs. 2–4 months in Fig. 5D), absence and presence of ASO injection, or lot-to-lot variation of the ELISA kits; we commented on this in the figure legend of Fig. 5D.

Regarding the substantial (~70%) decrease in the extracellular levels of glycine induced by the 1.3-fold increase in SLC6A20 protein levels may represent the consequence of continued action of SLC6A20A, although the driving force might be reduced as the concentration gradient gets lowered. The suggested potential reversal in the transporter activity of SLC6A20A in the brain of *Slc6a20a*^{-/-} mice might be possible, although this hypothesis should be directly and carefully tested in future studies with prior determination of glycine as well as proline concentrations in blood and brain compartments.

Point 3

The authors confirm "that the PTEN PB motif is required for synaptic translocation of PTEN during LTD" (Jurado S et al., 2010), but not for LTP induced by theta-burst stimulation (Knafo S et al., 2016). In contrast, they found that high-frequency tetanic stimulation failed to produce the late phase of LTP in PTEN DC/DC mice. Because LTD and LTP-HFS are NMDA-dependent, the authors conclude that NMDA signaling is suppressed in this mutant mouse and then examine NMDA current and the glycine depletion hypothesis. However, theta-burst LTP, which is a more robust experimental paradigm to trigger stable LTP than high-frequency tetanic stimulation (Hoffman DA et al., 2002; Larson J and Munkácsy E, 2015), is also dependent on NMDAR activation (Larson J and Munkácsy E, 2015) and it is not clear why the authors bypass this fact that contradicts their interpretation.

→ We appreciate these comments, which correctly pointed out our misinterpretation. Indeed, TBS-LTP is more efficient in inducing LTP than HFS-LTP and also involves non-NMDAR components such as presynaptic mechanisms (Larson & Munkacsy, 2015).

Therefore, the difference between WT and *Pten*^{ΔC/ΔC} mice in NMDAR function might be dampened under TBS-LTP conditions, relative to HFS-LTP conditions. We commented it in Results.

Kim's group already described NMDA-dependent suppression of LTD and LTP in the analysis of the Shank^{-/-} knockout mice (Won H et al., 2012). Like here, the hypofunction of NMDA receptors was the immediate cause since the plasticity was restored by adding D-Cycloserine (Won H et al., 2012), although Shank inactivation is unlikely to alter the extracellular glycine level. Therefore, interfering with the PSD-95 may lower NMDA signaling by many mechanisms. For example, we can speculate that changing the distribution of synaptic vs. extrasynaptic NMDA receptors would shift the coagonist sensitivity of NMDA receptor from D-serine (synaptic) to glycine (extrasynaptic) (Papouin T et al., 2012), and thus provide similar change in plasticity.

→ This is an interesting idea. We have to point out, however, that Shank2 is tightly coupled with GKAP, PSD-95, and NMDARs at excitatory synapses and is known to cause changes in NMDAR currents as well as synaptic levels of NMDAR subunits (Schmeisser et al, 2012; Won et al, 2012). However, our data suggest that *Pten* C-terminal deletion does not affect synaptic levels of NMDAR subunits (**Fig. EV1H**). In addition, *Pten* is not tightly associated with postsynaptic multi-protein complexes, being recruited to synapses in a regulated manner during synaptic plasticity or Aβ-induced synaptic weakening, as reported previously (Jurado et al, 2010; Knafo et al, 2016). Moreover, our results indicate that glycine concentrations are correlated with NMDAR functions in multiple experimental conditions (untreated/ASO-treated *Pten*^{ΔC/ΔC} mice and *Slc6a20a*^{-/-} mice). Therefore, it is unlikely that the suggested changes occur in *Pten*^{ΔC/ΔC} mice. We mentioned these aspects in Discussion.

Selection of minor problems.

It is hard to accept that a 25% reduction in the number of excitatory synapses does not affect the input/output relationship. What does mean synapses survival for fiber recruitment?

→ We should have been clear about this. We meant by the input/output relationship that the magnitude of postsynaptic responses (fEPSC amplitudes) are correlated with the strength of action potentials arriving at nerve terminals induced by axonal stimulation (fiber volley amplitudes). If we plotted, fEPSC amplitudes against the strength of the stimulation at presynaptic axon fibers (stimulus strength) rather than fiber volley amplitude, we could have seen a difference in the input-output relationship between genotypes. However, we did not use this parameter (stimulus strength) because it is a highly variable parameter depending on how close the electrode tip is to axon fibers being stimulated. We clarified this in the main text by changing "...as shown by input-output curve of fEPSP amplitudes plotted against fiber volley amplitudes" to "...as shown by input-output curve of fEPSP amplitudes plotted against fiber volley amplitudes (the strength of action potentials arriving at nerve terminals)".

The repeated usage (35) of the adjective strong, strongly, stronger to describe each

results is often not supported by the actual data.

Page 8: "strong decrease" is 25% reduction of synaptic element without any consequence

Page 8: LTP was "strongly suppressed": suppressed or strongly reduced would make more sense.

Page 8: again, LTD was "strongly suppressed."

Page 9: NMDA was "strongly reduced"

Page 11 "the strongly increased expression of Slc6a20a": Here, strongly refers to a 1.3 fold overexpression, which seems marginal. Seven occurrences of strong are found in page 11.

Page 12 "deletion of the PTEN PB motif induces a strong increase in transmembrane transport-related functions".

Page 12 "slc6a20a gene showed a strong increase in the expression"

Page 12 "exhibiting fairly strong expression in astrocyte".

etc

→ We corrected them in the text.

References from the reviewer

Bröer S, Bailey CG, Kowalczyk S, Ng C, Vanslambrouck JM, Rodgers H, Auray-Blais C, Cavanaugh JA, et al. (2008), Iminoglycinuria and hyperglycinuria are discrete human phenotypes resulting from complex mutations in proline and glycine transporters. *J Clin Invest* 118:3881-3892.

Dahlin A, Royall J, Hohmann JG, Wang J (2009), Expression Profiling of the Solute Carrier Gene Family in the Mouse Brain. *J Pharmacol Exp Ther* 329:558-570.

Dörrbaum AR, Kochen L, Langer JD, Schuman EM (2018), Local and global influences on protein turnover in neurons and glia. *Elife* 7:e34202.

Fornasiero EF, Mandad S, Wildhagen H, Alevra M, Rammner B, Keihani S, Opazo F, Urban I, et al. (2018), Precisely measured protein lifetimes in the mouse brain reveal differences across tissues and subcellular fractions. *Nat Commun* 9:4230.

He L, Vanlandewijck M, Raschperger E, Mäe MA, Jung B, Lebouvier T, Ando K, Hofmann J, et al. (2016), Analysis of the brain mural cell transcriptome. *Sci Rep-uk* 6:35108.

Hoffman DA, Sprengel R, Sakmann B (2002), Molecular dissection of hippocampal theta-burst pairing potentiation. *Proc National Acad Sci* 99:7740-7745.

Jurado S, Benoist M, Lario A, Knafo S, Petrok CN, Esteban JA (2010), PTEN is recruited to the postsynaptic terminal for NMDA receptor-dependent long-term depression. *The EMBO Journal* 29:2827-2840.

Kadakkuzha BM, Liu X-A, McCrate J, Shankar G, Rizzo V, Afinogenova A, Young B, Fallahi M, et al. (2015), Transcriptome analyses of adult mouse brain reveal enrichment of lncRNAs in specific brain regions and neuronal populations. *Front Cell Neurosci* 9:63.

Knafo S, Sánchez-Puelles C, Palomer E, Delgado I, Draffin JE, Mingo J, Wahle T, Kaleka K, et al. (2016), PTEN recruitment controls synaptic and cognitive function in Alzheimer's models. *Nat Neurosci* 19:443-453.

Kowalczyk S, Bröer A, Munzinger M, Tietze N, Klingel K, Bröer S (2005), Molecular cloning of the mouse IMINO system: an Na⁺- and Cl⁻-dependent proline transporter. *Biochemical Journal* 386:417-422.

Larson J, Munkácsy E (2015), Theta-burst LTP. *Brain Res* 1621:38-50.

Papouin T, Ladépêche L, Ruel J, Sacchi S, Labasque M, Hanini M, Groc L, Pollegioni L, et al. (2012), Synaptic and extrasynaptic NMDA receptors are gated by different endogenous coagonists. *Cell* 150:633-646.

Takanaga H, Mackenzie B, Suzuki Y, Hediger MA (2005), Identification of Mammalian Proline Transporter SIT1 (SLC6A20) with Characteristics of Classical System Imino. *Journal of Biological Chemistry* 280:8974-8984.

Won H, Lee H-R, Gee HY, Mah W, Kim J-I, Lee J, Ha S, Chung C, et al. (2012), Autistic-like social behaviour in Shank2-mutant mice improved by restoring NMDA receptor function. *Nature* 486:261-265.

References from our responses

Broer S, Bailey CG, Kowalczyk S, Ng C, Vanslambrouck JM, Rodgers H, Auray-Blais C, Cavanaugh JA, Broer A, Rasko JE (2008) Iminoglycinuria and hyperglycinuria are discrete human phenotypes resulting from complex mutations in proline and glycine transporters. *The Journal of clinical investigation* 118: 3881-3892

Clare JJ, Trezise DJ (2006) Advantages and Disadvantages of *Xenopus* Oocytes. In *Expression and Analysis of Recombinant Ions Channels*, Clare JJ, Trezise DJ (eds).

Clark JA, Amara SG (1994) Stable expression of a neuronal gamma-aminobutyric acid transporter, GAT-3, in mammalian cells demonstrates unique pharmacological properties and ion dependence. *Molecular pharmacology* 46: 550-557

Clark JA, Deutch AY, Gallipoli PZ, Amara SG (1992) Functional expression and CNS distribution of a beta-alanine-sensitive neuronal GABA transporter. *Neuron* 9: 337-348

Hall CN, Reynell C, Gesslein B, Hamilton NB, Mishra A, Sutherland BA, O'Farrell FM, Buchan AM, Lauritzen M, Attwell D (2014) Capillary pericytes regulate cerebral blood flow in health and disease. *Nature* 508: 55-60

He L, Vanlandewijck M, Raschperger E, Andaloussi Mae M, Jung B, Lebouvier T, Ando K, Hofmann J, Keller A, Betsholtz C (2016) Analysis of the brain mural cell transcriptome. *Scientific reports* 6: 35108

Jurado S, Benoist M, Lario A, Knafo S, Petrok CN, Esteban JA (2010) PTEN is recruited to the postsynaptic terminal for NMDA receptor-dependent long-term depression. *The EMBO journal* 29: 2827-2840

Knafo S, Sanchez-Puelles C, Palomer E, Delgado I, Draffin JE, Mingo J, Wahle T, Kaleka K, Mou L, Pereda-Perez I et al (2016) PTEN recruitment controls synaptic and cognitive function in Alzheimer's models. *Nature neuroscience* 19: 443-453

Larson J, Munkacsy E (2015) Theta-burst LTP. *Brain research* 1621: 38-50

Liu QR, Lopez-Corcuera B, Mandiyan S, Nelson H, Nelson N (1993) Molecular characterization of four pharmacologically distinct gamma-aminobutyric acid transporters in mouse brain [corrected]. *The Journal of biological chemistry* 268: 2106-2112

Palacin M, Estevez R, Bertran J, Zorzano A (1998) Molecular biology of mammalian plasma membrane amino acid transporters. *Physiological reviews* 78: 969-1054

Schmeisser MJ, Ey E, Wegener S, Bockmann J, Stempel V, Kuebler A, Janssen AL, Bourgeron T, Gundelfinger ED, Boeckers TM (2012) Autistic-like behaviours and hyperactivity in mice lacking ProSAP1/Shank2. *Nature*

Vanlandewijck M, He L, Mae MA, Andrae J, Ando K, Del Gaudio F, Nahar K, Lebouvier T, Lavina B, Gouveia L et al (2018) A molecular atlas of cell types and zonation in the brain vasculature. *Nature* 554: 475-480

Won H, Lee HR, Gee HY, Mah W, Kim JI, Lee J, Ha S, Chung C, Jung ES, Cho YS et al (2012) Autistic-like social behaviour in Shank2-mutant mice improved by restoring NMDA receptor function. *Nature* 486: 261-265

Thank you for the submission of your revised manuscript to EMBO Molecular Medicine. We have now received the enclosed report from two referees who were asked to re-assess it. As you will see the referees are now supportive and I am pleased to inform you that we will be able to accept your manuscript pending the following amendments:

***** Reviewer's comments *****

Referee #1 (Comments on Novelty/Model System for Author):

Most experiments have used mouse brain tissues, which is directly relevant to the studied topic.

Referee #1 (Remarks for Author):

The manuscript by Bae et al has provided multiple lines of evidence revealing SLC6A20A as a novel glycine transporter in the brain. SLC6A20A is abnormally increased in mice carrying a mutant PTEN protein, leading to the reduced extracellular level of brain glycine and decreased NMDAR currents. Elevating glycine level by SLC6A20 knockdown normalizes NMDAR currents and repetitive climbing behavior, suggesting that SLC6A20 inhibition has therapeutic potential for brain disorders involving NMDAR hypofunction. Authors have fully addressed prior concerns and improved the strength of the main conclusions. The 14 supplementary figures need to be labelled with Sup. Fig. ? to make it easier to track, and the supplementary tables should not be in the combined pdf file for the paper (now 1389 pages).

The authors have made all requested editorial changes.

We are pleased to inform you that your manuscript is accepted for publication and is now being sent to our publisher to be included in the next available issue of EMBO Molecular Medicine.

Corresponding Author Name: Eunjoon Kim
Journal Submitted to: EMBO Molecular Medicine
Manuscript Number: EMM-2020-12632-V2